

# Influence of temperature fluctuations on equilibrium ice sheet volume

Troels Bøgeholm Mikkelsen[1], Aslak Grinsted[1], and Peter Ditlevsen[1]

[1]Centre for Ice and Climate, Niels Bohr Institute, Juliane Maries Vej 30, DK-2100 Copenhagen Ø

*Correspondence to:* Troels Mikkelsen (bogeholm@nbi.ku.dk)

**Abstract.** Forecasting the future sea level relies on accurate modeling of the response of the Greenland and Antarctic ice sheets to changing temperatures. We show why the steady state of an ice sheet is biased toward larger size if the interannual weather generated fluctuations in temperature are not taken into account in numerical modeling of the ice sheet. We illustrate this in a simple ice sheet model. This bias could, if not taken into account, imply that the risk of collapse in a given climate change scenario is underestimated. We estimate that the effect of temperature variability on the surface mass balance of the Greenland Ice Sheet in recent ensemble forecasting should be adjusted downward by approximately 13 percent of the present day observed value, if assuming a 2 degree warming. Many predicted scenarios of the future climate show an increased variability in temperature over much of the Earth. In light of our findings it is important to gauge the extent to which this increased variability will further influence the mass balance of the ice sheets.

## 1 Introduction

Using coupled climate and ice sheet models, long time forecasting is often made computationally feasible by running a climate model for one or more years and then repeatedly applying the climate (or the surface mass balance computed from it) to an ice sheet model (Kageyama et al., 2004; Gregory et al., 2012; Ziemen et al., 2014). Some studies (e.g. Kageyama et al. (2004)) compute the surface mass balance from a climatology. The present analysis shows that computing the surface mass balance from a climatology can result in a bias towards a larger ice sheet size, if the surface mass balance is estimated assuming a yearly averaged temperature.

Ice sheet modeling and evidence from paleoclimatic records indicate that ice sheets display a hysteresis response to climate forcing (Abe-Ouchi et al., 2013; Robinson et al., 2012). There is a critical threshold in temperature, a tipping point, beyond which an ice sheet becomes unsustainable. This is a generic saddle-node bifurcation point, estimated by Robinson et al. (2012) to be reached for the Greenland Ice Sheet (GrIS) at a global warming of $+1.6°C$ $(0.8°C – 3.2°C)$ above preindustrial.

Several recent studies suggest that parts of the West Antarctic Ice Sheet (WAIS) may already have been destabilized (Favier et al., 2014; Joughin et al., 2014; Rignot et al., 2014; Mouginot et al., 2014; Seroussi et al., 2014). Other studies find that East Antarctica may be more vulnerable to warming than previously thought (Mengel and Levermann, 2014; Greenbaum et al., 2015; Sun et al., 2014; Pollard et al., 2015; Fogwill et al., 2014). There is a growing concern for a considerable risk of a marine





ice-sheet instability of the WAIS may lead to a substantial sea level rise contribution already this century (Bamber and Aspinall, 2013).

Paleoclimatic records show a nonlinear relationship between temperature increase and sea level rise consistent with the threshold behavior of ice sheets, predicted by modeling studies. Gasson et al. (2012); Foster and Rohling (2012) find that even a moderate global warming of $+2°C$ or $CO_2$ levels of 400 ppm is associated with a likely long-term sea level rise of more than 9 m. This is consistent with evidence from the last interglacial which points toward a collapse of the WAIS (Kopp et al., 2009; Dahl-Jensen et al., 2013; Strugnell et al., 2012). Likewise there is evidence for at least one substantial deglaciation period in Greenland having occurred during the past 1.1 million years (Blard et al., 2016; Bierman et al., 2016; Schaefer et al., 2016).

The greenhouse gas concentrations and intense warming in high-end scenarios such as ECP8.5 (Extended Concentration Pathways, extension of Representative Concentration Pathways beyond 2100) (Meinshausen et al., 2011) correspond to an ice-free planet in the paleoclimatic record (Gasson et al., 2012; Foster and Rohling, 2012) which evidence suggests was the case until approximately 35 million years ago (Ruddiman, 2014).

Observations, paleoclimatic records and model studies indicate a real risk of ice sheet collapse for realistic future scenarios global warming. A substantial part of WAIS may already be committed to collapse. The threshold for GrIS is estimated to be passed in ECP4.5 and ECP6, and even total deglaciation is within reach of the ECP8.5 scenario. The complete loss of the Greenland –, the West Antarctic –, and the East Antarctic ice sheets would raise global sea levels by 7.4 m, 4.3 m, and 53 m respectively, excluding any solid earth rebound effects that would take place during ice sheet decay (Bamber et al., 2013; Fretwell et al., 2013). The risk that global warming might exceed the tipping points of ice sheet stability pose an existential threat to low lying coastal nations. Estimating how close each ice sheet is to a tipping point is thus critically important.

The stability of ice sheets is typically investigated by imposing a constant climate forcing and then letting the ice sheet model reach equilibrium (Robinson et al., 2012; Solgaard and Langen, 2012; Huybrechts and de Wolde, 1999). The hysteresis curve, and collapse thresholds are then traced out by repeating these experiments for a range of temperatures and starting from ice free conditions. However, this approach disregards the effects of interannual variability.

In the classical study of the effect of asynchronous coupling by Pollard et al. (1990) it was noticed that a stochastic forcing in an ice sheet model results in a smaller ice sheet in comparison to a constant constant forcing. Here we show how variability in forcing changes the expected mass balance of an ice sheet. We develop a general theoretical framework for how forcing variability impact the expected response in a model that exhibits a non-linear response. We illustrate the importance using a minimal model of how Greenland surface mass balance responds to temperature fluctuations. The simple model is also used to assess the bias adjustments needed in model studies when constant forcing is applied.

Though some studies implement full GCM coupling to the ice sheet model, or have some mixed approaches (Ridley et al., 2005) (Gregory and Huybrechts, 2006), the computational demand of the GCM could come at an expense for the resolution of the ice sheet flow model. The results presented here show explicitly how to account for the effect of unresolved temperature variability.





Previous studies of natural variability in the context of ice sheets include Fyke et al. (2014), who find that the variability of the GrIS surface mass balance will increase in a warmer climate due to increased ablation area, and Roe and O'Neal (2005) who find that large fluctuations in glacier extent can be driven by natural, fast fluctuations in climate.

That the SMB of an ice sheet model is nonlinear is well known. Ridley et al. (2010) specifically avoid using monthly
climatologies in order to include the effect of interannual variability in their study. Seguinot (2013) shows how simplifying assumptions (in general leading to lower temperature variability) in a positive degree day (PDD) scheme leads to errors. Fettweis et al. (2013, see Fig. 6h) investigate the GrIS SMB simulated by regional climate models (RCM) as a function of mean surface temperature from general circulation models (GCM). Our contribution is a quantification of this effect, and an estimate of the necessary bias correction in long term ice sheet simulations.

Sub-annual temperature variability in the context of positive degree-day (PDD) models is investigated in eg. Hock (2003); Seguinot (2013); Wake and Marshall (2015). PDD models connect surface melting and air temperature, and are used extensively due to their simplicity and wide availability of air temperature data (Hock, 2003). Seguinot (2013) compares Greenland SMB calculated from four different annual PDD formulations with a reference SMB calculated from a PDD scheme using a monthly climatology and deviations from an long-term interannual mean. At the scale of sub-annual climatology, there are
large uncertainties as the estimates of the SMB differs significantly among the formulations, highlighting the need to accurately model both spatial and temporal variability. These findings are built upon by Wake and Marshall (2015) who find that the standard deviation of monthly average temperature may be represented as a quadratic function of monthly average temperature. In the present study we are concerned with interannual variability and expect our results to apply independently of the chosen SMB model.

In Section 2 we derive an analytical relationship between the magnitude of temperature fluctuations assuming a simple relationship between the mass balance, temperature and ice sheet volume. This relationship is shown to hold using a simple ice sheet model (including a surface mass balance model) in Section 3, and in Section 4 we estimate the consequences of temperature fluctuations on a recent long term ice sheet study, assuming the effect we present here is not already accounted for. The limitations of this approach, as well as further possible applications, is discussed in Section 5.

## 2  The Mass Balance of an Ice Sheet

### 2.1  A Minimal Ice Sheet Model

We consider a simple ice sheet model introduced by Oerlemans (2003) hereafter denoted *Oer03*. This model describes the essential dynamics of an ice sheet initiated from a mountain glacier. It assumes an axially symmetric ice sheet resting on a bed that slopes linearly downwards from the center. The ice is modeled as a perfectly plastic material, and the ice sheet is coupled
to the surrounding climate by adjusting the height of the equilibrium line $h_{Eq}$ (Oerlemans, 2008):

$$h_{Eq} = h_{E,0} + (T - \bar{T}) \cdot 1000/6.5. \tag{1}$$



Equation 1 represents an increase of the equilibrium line altitude of roughly 154 m °C$^{-1}$. The influence of $h_{Eq}$ on the specific balance $B$ is illustrated in Fig. (1). It should be noted that this simple relationship does not capture situations where the SMB may *increase* with increasing temperature, as discussed in Section 5.

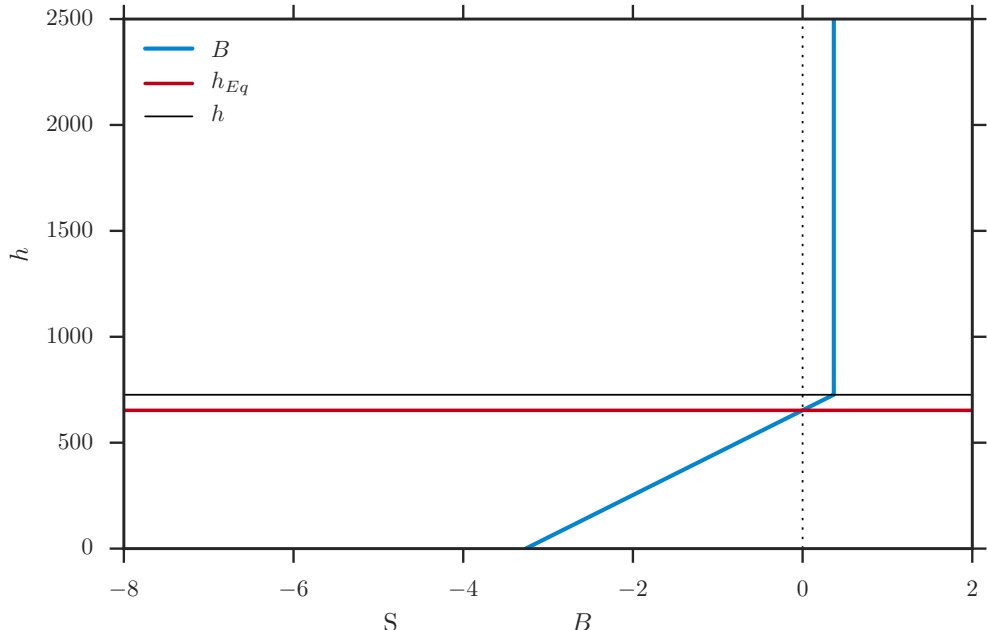

**Figure 1.** Specific balance $B$ for $\overline{T} = 0$ from Eq. (1) using the parameters in Table 1 and Eqs. (3–4) of the supplement. $h_{Eq}$ denotes the equilibrium line. The specific balance is constant above the runoff line $h_r$ (Supplementing Information, Eq. (4)), and the balance gradient is constant below $h_r$ (Oerlemans, 2003).

The model is chosen for its simplicity, thus it is not accurately modeling a specific ice sheet; the two main reasons for choosing it for our analysis are: 1) The simplicity of Oer03 allows the analytical approach detailed below and 2) the Oer03 model shows the same functional relationship between surface mass balance (SMB) and temperature as has been found for regional climate models (RCM) for a range of temperature scenarios (Fettweis et al., 2013). The change in volume or mass of the ice sheet depends on the balance between accumulation, ablation and ice sheet discharge which in turn depends on both the interplay between the fluctuating temperature and the state of the ice sheet itself.

Before proceeding with the simple model, we investigate the effect of interannual temperature fluctuations by considering the ice sheet as a simple dynamical system. Assume the mass balance of the ice sheet to depend only on the volume $V$ itself and a single time-varying mean temperature over the ice sheet, $T$; thus all components of the mass budget are uniquely determined by temperature and volume. This is a vast simplification but sufficient to illuminate the essential dynamical effect we consider in this paper. Denoting the mass balance (change in ice sheet volume) as $\dot{V}$,

$$\dot{V} = f(T, V), \tag{2}$$




where $f(V, T)$ is some non-linear function. The (stable) fixed point, $f(T, V) = 0$ corresponds to a balance between loss and gain in the ice volume. This is in general an implicit equation to determine the steady state volume $V_0(T)$ as a function of temperature, such that $f(V_0(T), T) = 0$.

However, the fixed point is not identical to the statistically steady state volume with a temporally fluctuating temperature $T_t = T(t)$ with expectation value $\langle T_t \rangle = \overline{T}$. A numerical integration to equilibrium of an ice sheet model with and without interannual fluctuating temperature shows that in steady state the ice sheet volume $V_t$ will fluctuate around $\langle V_t \rangle = \overline{V}$ where $\overline{V}$ is systematically smaller than the corresponding $V_0(T)$ (Fig. 2).

Since the temperature $T_t$ – and thus the ice sheet volume $V_t$ – is a stochastic variable the following will characterize an equilibrium state:

$$\langle f(T_t, V_t) \rangle = 0. \tag{3}$$

To calculate $\overline{V}$ we perform a Taylor expansion of Eq. (3) around the – presently unknown – steady state $(\overline{T}, \overline{V})$ and calculate the mean volume $\overline{V}$. We use the notation $f_T := \frac{\partial f}{\partial T}$, $f_{TV} := \frac{\partial^2 f}{\partial T \partial V}$, etc. Furthermore, $f^0 := f(\overline{T}, \overline{V})$, $f_T^0 := \frac{\partial f}{\partial T}(T, V)\Big|_{(\overline{T}, \overline{V})}$ etc. We then get:

$$\langle f(T_t, V_t) \rangle = f^0 + \langle T_t - \overline{T} \rangle f_T^0 + \langle V_t - \overline{V} \rangle f_V^0 + \frac{1}{2} \langle (T_t - \overline{T})^2 \rangle f_{TT}^0$$
$$+ \frac{1}{2} \langle (V_t - \overline{V})^2 \rangle f_{VV}^0 + \langle (T_t - \overline{T})(V_t - \overline{V}) \rangle f_{TV}^0 + \mathcal{O}(3), \tag{4}$$

where $\mathcal{O}(3)$ represents higher order terms.

We can simplify Eq. (4) considerably: First note that since $\overline{T}$ is the expectation value of $T_t$ we have $\langle T_t - \overline{T} \rangle = \langle T_t \rangle - \overline{T} = \overline{T} - \overline{T} = 0$ and with the same argument $\langle V_t - \overline{V} \rangle = 0$. The quantity $\langle (T_t - \overline{T})^2 \rangle$ is the variance of the fluctuating temperature – we will assume this is known in simulations and substitute $\langle (T_t - \overline{T})^2 \rangle = \sigma_T^2$. Since the temperature variations are small with respect to the mean and has a symmetric distribution we may neglect higher order terms in Eq. (4) (Rodriguez and Tuckwell, 1996). We are left with:

$$\langle f(T_t, V_t) \rangle \approx f^0 + \frac{\sigma_T^2}{2} f_{TT}^0$$
$$+ \frac{1}{2} \langle (V_t - \overline{V})^2 \rangle f_{VV}^0 + \langle (T_t - \overline{T})(V_t - \overline{V}) \rangle f_{TV}^0. \tag{5}$$

We have evaluated the last two terms in Eq. (5) numerically for the model presented in Section 3 and found that $\langle (V_t - \overline{V})^2 \rangle$ and $\langle (T_t - \overline{T})(V_t - \overline{V}) \rangle$ tend to zero (supplementing information) – neglecting the last two terms, Eq. (5) reduces to

$$\langle f(T_t, V_t) \rangle \approx f^0 + \frac{\sigma_T^2}{2} f_{TT}^0. \tag{6}$$

Equation (6) is the main observation in this work. We shall in the following estimate the implications of this result on realistic asynchronously coupled state-of-the-art ice sheet climate model simulations. As $\langle f(T_t, V_t) \rangle = 0$ at the steady state it can be seen from Eq. (6) that

$$0 = f^0 + \frac{\sigma_T^2}{2} f_{TT}^0 \Rightarrow$$
$$f^0 = -\frac{\sigma_T^2}{2} f_{TT}^0 > 0 \tag{7}$$





since $f^0_{TT} < 0$ – this negative curvature of $f^0$ is the nonlinear effect causing the bias. $V_0(T)$ is the stable fixed point; $f(V_0(T), T) = 0$, thus $f(V, T) > 0$ for $V < V_0$ and $f(V, T) < 0$ for $V > V_0$. This together with Eq. (7) implies that $\overline{V} < V_0$; that is, positive temperature anomaly increases the mass loss more than what can be compensated by an equally large negative anomaly (van de Wal and Oerlemans, 1994).

## 3 Ice Sheet Simulations

### 3.1 Fluctuating Temperatures

To generate an ensemble of volume simulations we use time series $T_t$ comparable to the observed temperatures over Greenland between year 1851 and 2011. For this we use the AR(1)-process (Hasselmann, 1976; Frankignoul and Hasselmann, 1977; von Storch and Zwiers, 2003; Mudelsee, 2010):

$$T_{t+1} = \overline{T} + a \times (T_t - \overline{T}) + \sigma_{AR} W_t. \tag{8}$$

The parameters $(a, \sigma^2_{AR})$ were obtained by fitting Eq. (8) to the observed annual mean temperatures over Greenland between year 1851 and 2011 (supplementing information). We obtain $(a, \sigma^2_{AR}) = (0.67, 0.85)$ thus the process Eq. (8) has variance (Box et al., 2008) $\sigma^2_T = \sigma^2_{AR}/(1-a^2) = 1.54$ K$^2$ comparable to the observed annual mean temperature variance over Greenland, $\sigma^2_{T,obs} = 1.55$ K$^2$.

As we quantify the effect of interannual stochastic variability we use yearly averaged temperatures, consistent with the formulation of the Oer03 model (cf. Table 1 of the Supplementing Information). We find time step size of one year to be sufficient for integrating the Oer03-model (supplementing information); thus $T_{t+1}$ in Eq. (8) represents the temperature one year after $T_t$.

To find the steady state volume we run the Oer03-model forward long enough for the ice sheet to reach equilibrium, with and without fluctuating temperatures. The results of this procedure are shown in Fig. 2 (left) where it is clearly seen that the steady state volume is lower for simulations with fluctuating temperatures than with constant temperature. We emphasize that the fluctuating temperature time series $\{T_t\}$ have as mean the constant temperature, $\langle T_t \rangle = \overline{T}$ so that the differences are due only to the annual temperature fluctuation.

In Fig. 2 (right) the effect of temperature fluctuations is shown in the $(T, V)$-plane: the markers "+" are steady states of numerical simulations with constant temperature, while the circles represent ensemble averages of simulations with fluctuating temperatures. It is evident that temperature fluctuations decrease the steady state ice volume. The yellow curve in Fig. 2 (right) was calculated using Eq. (6) and gives a good agreement with the results from ensemble simulations.

In order to illustrate the physics behind Eq. (6), consider values of the mass budget function $f$ for different ice sheet volumes $V$, shown in Fig. 3. The insert shows, for a particular value of $V$, how the steady state is influenced by fluctuating temperatures: the average mass budget of a colder year and a warmer year is less than the mass budget of a year with a temperature corresponding to the average of "cold" and "warm"; to put it another way: the increased SMB of a single anomalously cold year cannot balance the increased melt from an equally anomalously warm year (van de Wal and Oerlemans, 1994). In particular let



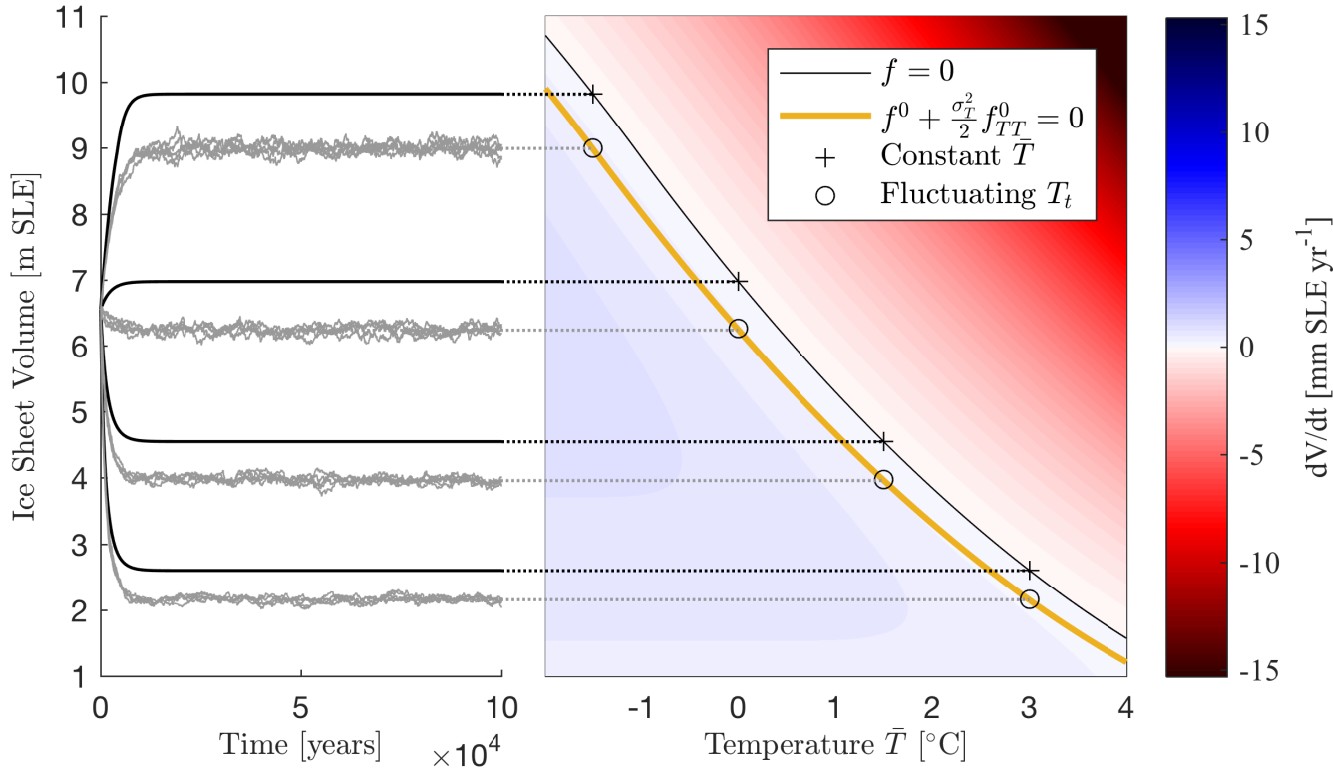

**Figure 2.** (Left) Simulations of the Oer03-model for $\overline{T} = -1.5, 0, 1.5$ and $3$. The black curves denote a constant temperature and the grey curves fluctuating temperatures generated with Eq. (8). (Right) The mass balance Eq. (2) for the Oer03-model in the $(T, V)$-plane. The black contour is the steady state $f = dV/dt = 0$. The markers represent the average of the numerical simulation with constant (+) and fluctuating (○) temperature seen on the left. Finally the yellow contour shows the approximation derived in in Eq. (6).

$T_c = \overline{T} - \sigma$ and $T_h = \overline{T} + \sigma$:

$$\frac{f(V, T_c) + f(V, T_h)}{2} < f\left(V, \frac{T_c + T_h}{2}\right), \tag{9}$$

which is consistent with $f_{TT}^0 < 0$ as shown in Eq. (7).

## 4   Consequences for Long Term Ice Sheet Simulations

Here we investigate the effect of accounting for fluctuating temperatures when running long time scale climate simulations. These can be either transient runs, scenarios with specified changing $CO_2$-forcing or equilibrium runs with specified constant forcing. Specifically we analyze the results of Robinson et al. (2012) where the long term stability of the GrIS is investigated. In this study, an ice sheet model is forced by the output of a regional climate model driven by the ERA40 climatology with a constant temperature anomaly applied, see Robinson et al. (2012) and Supplementary Information.




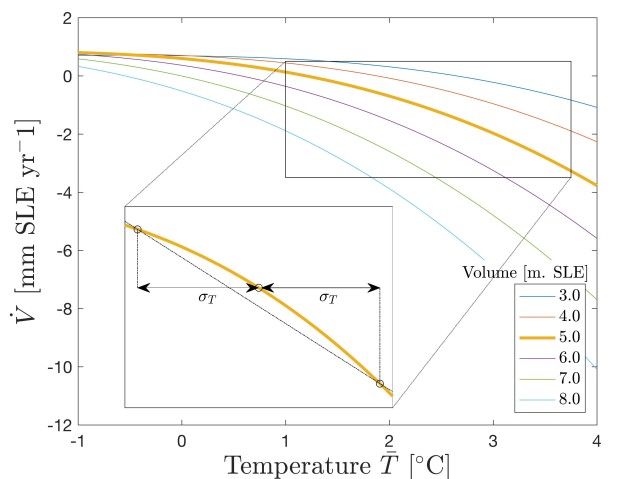
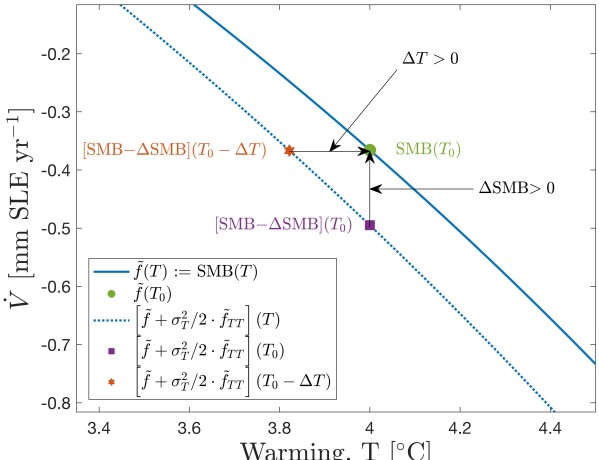

**Figure 3. Left:** Mass balance $\dot{V}$ of the ice sheet for different values of the total ice sheet ice volume $V$ in the Oer03-model. Similar to Fig. 2 but here we show $\dot{V}$ as a function of $\overline{T}$ for different total volumes $V$. **Insert, left:** The curvature of $\dot{V}(\overline{T})$ influences the steady state behavior – a cold year does not cancel out the effect of an equally warm year as shown in Eq. 9. The value of $\sigma_T$ is used for illustration and is given as the square root of the temperature variance, $\sigma_T = \sqrt{1.54\mathrm{K}^2} = 1.24\mathrm{K}$. Note the similarity of the $\dot{V}(\overline{T})$ found here to Fig. 6h in Fettweis et al. (2013). **Right:** Estimating the effect of fluctuating temperatures on GrIS projections. The full curve is obtained by fitting a third degree polynomial $\tilde{f}(T)$ to an $\mathrm{SMB}(T)$ from Robinson et al. (2012). The dotted line show the effect of temperature fluctuations obtained by applying Eq. (6). For a warming of $4°\mathrm{C}$ the green circle shows the SMB. $\Delta\mathrm{SMB}$ is obtained by applying Eq. (11) and represents the change in mass balance resulting from the temperature fluctuations. $-\Delta T$ is the temperature change required to negate this effect and is obtained implicitly from Eq. (12).

As parameters in ice sheet models are often tuned to best match the problem under investigation (eg., Muresan et al. (2016)), the ice sheet volume bias we describe may already be implicitly compensated. To estimate the size of the temperature fluctuation bias, we assume that this has not already been accounted for by parameter tuning.

Fettweis et al. (2013) compare the output of RCMs forced with multiple future climate scenarios and show that the effect of
5    rising temperature on the GrIS SMB is well described by a third degree polynomial (note the qualitative similarities between Fig. 3 in the present article and Fig. 6h in Fettweis et al. (2013)). Here we take the same approach. To the ensemble of simulations in Robinson et al. (2012) we fit third degree polynomials to the SMB as a function of temperature at time $t = 200$ years (supplementing information) and obtain third degree polynomials in $T$:

$$\left\{ \tilde{f}_{ij}(T) \big| \tilde{f}_{ij}(T) = A_{ij}T^3 + B_{ij}T^2 + C_{ij}T + D_{ij} \right\} \tag{10}$$

10    where the indices $i$ and $j$ run over two separate parameters in the model that take 9 – respectively 11 –values (Robinson et al., 2012) so in total we have 99 unique polynomial fits. These polynomials are then used as a simple description of the mass balance function as a function of temperature, $\mathrm{SMB}_{ij}(T) = \tilde{f}_{ij}(T)$. Differentiating twice we obtain $\tilde{f}_{TT}(T) = 6AT + 2B$ (suppressing indices $i, j$ for clarity).

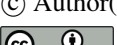



For all parameter pairs $(i, j)$ we evaluate $\tilde{f}(T)$ and $\tilde{f}(T) + (\sigma_T^2/2)\tilde{f}_{TT}(T)$ – this is shown in Fig. 3 (right) as the full and dotted lines, respectively.

To illustrate this approach we pick a specific temperature $T_0$. $\tilde{f}(T_0)$ is thus the SMB for a constant temperature and $\tilde{f}(T_0) + (\sigma_T^2/2)\tilde{f}_{TT}(T_0)$ represents the effect of letting the temperatures fluctuate. This procedure gives us an expression for $\Delta$SMB

$$
\begin{aligned}
\quad \Delta\text{SMB} = \quad & \tilde{f}(T_0) - \left[ \tilde{f}(T_0) + \frac{\sigma_T^2}{2}\tilde{f}_{TT}(T_0) \right] \\
= \quad & -\frac{\sigma_T^2}{2}\tilde{f}_{TT}(T_0) \quad\quad\quad\quad\quad\quad\quad\quad\quad\quad\quad\quad\quad\quad (11)
\end{aligned}
$$

where $\Delta$SMB is positive in accordance with Eq. (7). Next we find the temperature difference $\Delta T$ such that

$$
\tilde{f}(T_0 - \Delta T) + \frac{\sigma_T^2}{2}\tilde{f}_{TT}(T_0 - \Delta T) = \tilde{f}(T_0). \quad\quad\quad\quad\quad\quad\quad\quad\quad\quad\quad\quad (12)
$$

In this way $\Delta T$ is the *effective* temperature change resulting from considering fluctuating temperatures.

The results of applying the steps outlined above on the data from Robinson et al. (2012) are shown in Fig. 4 (see also supplementing information). The red curves in Fig. 4 shows the most likely $\Delta T$ and $\Delta$SMB; the grey curves are estimates for the $9 \times 11$ individual parameter values and the blue shade area represents the $95\%$ credibility region.

The warmings quoted in Robinson et al. (2012) are relative to the preindustrial period whereas the reported warming from the preindustrial period to the present day is estimated to $1°$C (Stocker et al., 2013, p. 78). Following the RCP45 scenario it is

*more likely than not* that Earth will experience a further warming of $2.0°$C (IPCC, 2013, p. 21) from today to the year 2100. Combing these numbers we arrive at a warming of $3.0°$C in the year 2100 relative to the preindustrial when considering the RCP45 scenario. For this value it is seen in Fig. 4 (top) that an additional $0.12°$C should be added to any constant warming term when considering simulations of the Greenland ice sheet, assuming the same temperature variance as in Section 3. Further, Fig. 4 (bottom) shows the most likely $\Delta$SMB resulting from temperature fluctuations at a $3°$C warming to be 30 Gt/y. To put

this number in context, consider Barletta et al. (2013) who report an average GrIS SMB of $-234 \pm 20$ Gt/y for the period 2003 to 2011.

Observe in Fig. (4) that $\Delta T$ goes to zero for low temperature anomalies and appears to saturate for higher temperature anomalies. In the framework presented here this can be explained by considering the SMB$(T)$-curves shown in Fig. (3) (left). For low temperature anomalies the SMB$(T)$ curve in Fig. 3 (left) is close to flat so the second derivative is small; this gives

a small contribution to $\Delta$SMB from Eq. (11). On the other hand, as the SMB$(T)$ curve in Fig. 3 (left) becomes progressively steeper, a correspondingly smaller $\Delta T$ in Eq. (12) is required to compensate for $\Delta$SMB.

The results above highlight that interannual temperature variability cannot be neglected in long term studies involving ice sheet models. The straightforward approach would be to simply include the expected temperature variability in a number of simulations followed by calculating the ensemble average. Conversely, one could calculate the effect of temperature variability

for a range of climate scenarios as a starting point for a following bias adjustment.





**Figure 4.** Maximum likelihood estimates of $\Delta T$ and $\Delta$SMB (red curves). The grey curves are estimates from individual simulations and the blue shaded area denotes 95% credibility regions.

## 5 Conclusions

### 5.1 Limitations of this study

When calculating the $\tilde{f}$'s in Eq. (10) and Eq. (11) we assume a constant volume in the data from Robinson et al. (2012), but in reality the relative variations are as large as 9.5% when considering all the warming temperatures shown in Fig. 4 (supplementing information). However to draw the conclusion about the consequences of a 3°C warming it is adequate to consider warmings less than 4°C and here the volume variation was less than 3% of the average. Neglecting variations in





volume does add uncertainty to our results, and it is not immediately clear to us how to quantify that uncertainty. Additionally, at time $t = 200$ years where we extracted the SMB data from the simulations in Robinson et al. (2012), the ice sheet models had not yet reached steady state; thus, expanding the analysis using a data set from ice sheet simulations in steady state would be desirable.

The temperature fluctuation is accounted for in most studies either explicitly (Ridley et al., 2010; Seguinot, 2013) or implicitly in the tuning of the surface mass balance scheme. Our result may be used to explicitly implement the contribution from the temperature fluctuations in the mass balance schemes before bias correcting due to other possible model deficiencies.

## 5.2   Conclusion and outlook

From a theoretical argument and by considering a minimal ice sheet model we have shown that fluctuating temperatures forcing
the ice sheet have an effect on the steady state volume of the ice sheet.

The effect is explained by the curvature, or second derivative, of the mass balance as a function of temperature. A negative curvature gives rise to nonlinear effects meaning that the average mass accumulation resulting from a cold year and a warm year in succession is less than the mass accumulation of two consecutive years having the average temperature of the "warm" and "cold" years.

Even though we considered a simple ice sheet model, the results are transferable to other more realistic models as long as the rather weak assumptions leading up to Eq. (6) hold; eg., models of sub-shelf melting, grounding line migration, and ice discharge respond very non-linearly to changes in ocean temperatures (Favier et al., 2014; Joughin et al., 2014; Seroussi et al., 2014; Mengel and Levermann, 2014; Pollard et al., 2015; Fogwill et al., 2014), thus it is critical to take variability into account for quantitative assessments.

The response of a real ice sheet to temperature increase is naturally much more complex than what can be described in a simple study such as the present paper. In a model study, Born and Nisancioglu (2012) observe mass loss acceleration of the Northeastern GrIS as a response to warming. This part of the GrIS experiences comparatively little precipitation and thus increasing melt is not compensated by increasing accumulation. However, the opposite has been shown to be the case for Antarctica. Frieler et al. (2015) show that increasing temperatures will *increase* Antarctic SMB at continental scales due to
increasing precipitation. This is an interesting special case of an accumulation dominated mass balance, where the curvature term in Eq. (6) has the opposite sign, thus an underestimated temperature fluctuation would lead to an underestimation of the growth of the ice sheet.

We have evaluated the consequences of the temperature fluctuation bias on long-term GrIS simulations and found that, if the full effects are taken into account with no further modifications, a significant *effective* temperature change would be required
for an unbiased estimation of the equilibrium ice volume.

## 6   Code availability

The code for this study is available upon request to the corresponding author.

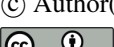



## 7  Data availability

Data used in this study was obtained from the authors of Robinson et al. (2012).

*Author contributions.*  TBM, AG and PD designed the study. TBM performed the data analysis. TBM, AG and PD wrote the article.

*Competing interests.*  The authors declare no conflict of interest.

5   *Acknowledgements.*  We would like to thank Johannes Oerlemans for providing the original code for his model and Alexander Robinson for being very helpful in providing the data from their study.

   Furthermore we are very thankful for valuable comment from the anonymous reviewers.

   This work is part of the Dynamical Systems Interdisciplinary Network (DSIN) – T.M. was financially supported by the Centre for Ice and Climate and the DSIN, both University of Copenhagen.



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
