# Peer review of "Influence of temperature fluctuations on equilibrium ice sheet volume"

_The Cryosphere, 2017_

## Referee Comment (RC1) · X. Fettweis (Referee) · 18 May 2017

This paper highlights the impact of taking into account the temperature variability in future projections of the GrIS SMB using a simple idealized model. As shown by Fettweis et al. (2013) and well mentioned in this paper, the temperature dependence of melt is not linear while precip increases linearly with temperature. Therefore, forcing a model with a mean climate or a climate resolving the interannual variability will be different. But, don't forget that such effect is explicitly taken into account in all of the simulations forced by GCMs. However, in the idealized models, this effect is often neglected and this paper evaluates the additional mass loss if the interannual variability is taken into account. This (technical) paper is well written, fits well with TC and deserves to be published.

[Figure]

However, we can not forget that the results presented here (13% of additional mass loss) are only valid for

- idealized models.

- for GrIS (where the future changes will be driven by melt increase) and not Antarctica (where the future changes will be driven by precip changes).

Therefore, any references to Antarctica in Abstract and Conclusion should be avoid at maximum (e.g. lines 23-27, pg 11) as the Antarctica SMB is, in a 1st order, linearly temperature dependent and the fact that this effect is already explicitly taken into account in all of the more complex/realistic GCMs forced simulations should be more clearly stated into Abstract and Conclusion (e.g. lines 15-19, pg 11).

Finally, the fact that the interannual variability of temperature could likely be not the dominant missing process (vs positive feedbacks) in idealized models should clearly be mentioned into the conclusion.

---

## Referee Comment (RC2) · G. Roe (Referee) · 28 May 2017

From the accompanying materials, it is suggested that the paper has undergone several rounds of review already. The editor has already supplied an impressive round of comments to which the authors have responded.

The principle result is one that is important to recognize in all nonlinear dynamic models: when models are calibrated to time-mean inputs, there will be bias in the model coefficients because nonlinearities act to rectify variability in the forcing. In my experience this basic point is not as widely appreciated as it should be in glaciology.

The results suggest that the effect is significant enough for ice sheets the size of Greenland and Antarctica, that the issue must be accounted when making future model projections. I think it is worth making the point in the context of ice sheets, and that the

result should be published.

I have three main comments and criticisms.

1. I question how important this effect is relative to other uncertainties. While the point is worth making, the size of the effect the authors find is hardly the rate-limiting uncertainty in ice-sheet projections, or in establishing the likelihood of, or proximity to, tipping points. The authors own calculations suggest the effect of variance is the same as changing the mean temperature by 0.12K. This is obviously very small compared to the spread of uncertainty in model projections of future climate change, polar amplification, and the parameterization of ablation. I think a revised manuscript should discuss the results in relation to other uncertainties; and the asserted importance in the abstract and introduction might be dialed down a bit.

2. The physical reason for the nonlinearity should be clearly described. As of now there is almost no explanation, it is presented as a model fact, and only recent papers are cited. A reader will likely crave having a physical reason provided.

That the mass balance should be nonlinear has long been known. It is implicit in the ELA sensitivities derived sixty years ago in Wertman (J Glac, 1960, 1963, Science 1976). The reason is also explicitly derived in Roe and Lindzen (Clim Dyn, 2001), and likely earlier and elsewhere. The ablation rate scales as temperature, and the ablation area scale as $\sim T^2$ because of the characteristic parabolic profile of ice sheets; giving a roughly cubic dependency for total ablation. There is also a smaller, but nontrivial effect, that the length of the melt season changes with T. Note the degree of nonlinearity will be different for the plastic-rheology profiles the authors use in the Oer03, from the dynamic ice model used in Robinson et al., so there is an internal inconsistency in the results presented here. (The degree of nonlinearity is larger for a shallow-ice rheology than it is for plastic-rheology ice sheet, by an increase of approximately one in the exponent.)

I don't understand why the authors did not use the temperature nonlinearity that is di-
rectly represented in the Oer03 model, and instead calibrated to a completely different model set-up from Robinson et al. (the former is an axisymmetric ice sheet and climate, the latter is a realistic Greenland). In Oer03 the ELA is directly specified in terms of temperature, and the geometric nonlinearity in the ablation-temperature relation certainly exists in Oer03. It would be a more self-consistent estimate of the effect, and certainly worth comparing with the extrapolations from Robinson et al.

If the authors have other mechanisms they have diagnosed or have speculations about, those should be given too. Otherwise it can be frustrating to read about an effect whose cause is not explained.

3. The application of stochastic climate variability.

The authors represent stochastic variability by applying AR(1) red noise in annual-mean temperatures.

Applying stochastic variability to the annual mean temperature is likely wrong. Annual-mean anomalies are the result of much larger stochastic variation in seasonal temperatures (seasonal fluctuations are $\sim$sqrt(4) larger than annual mean. A model will fail to emulate realistic mass balance anomalies without accounting for theses larger seasonal fluctuations that actually drive the ablation budget.

p5L9 "(AR2 AR10 ) = (0.67, 0.85)" What is the persistence timescale implied by this coefficient? For AR(1) tau = 1yr/(1-a) giving tau $\sim$3yrs. The uninitiated reader has no idea what the point of AR(1) is, and why it is important to use, so more explanation is needed. The persistence in the annual-mean anomalies are not going to be the same as the melt-season anomalies, which is important to account for. The ablation anomaly is due to the melt-season temperatures, and the melt-season persistence timescale is typically less than annual-mean.

Furthermore, for this to be rigorous, some kind of criterion should be used to evaluate whether AR(1) is a sufficient, self-consistent, and parsimonious description of the data.

There are various methods for establishing this (e.g.., Akaike information criteria, etc.), but none are referred to, so a reader has no idea of the necessity of this fit, or if any such estimate was performed. The data should be detrended before fitting, and the residuals should be tested for any remaining autocorrelation. I did not find either of these mentioned in the paper or supplementary.

Just by eye, the temperature time series shown in the supplementary looks a bit questionable between 1850 and 1900. I would recommend subsampling the data, and using other datasets (preferably instrumental records) to see how stable the estimates are.

Finally, AR(1) is a somewhat limited representation of climatic persistence. The spectrum flattens out at periods longer than 2*pi*tau, and so there is no persistence at multidecadal and centennial scales, in contrast to a power-law representation, for instance. The nature of climatic persistence at low frequencies is debated, but alternative representations would have important implications for these results and affect the answers quantitatively, so some discussion would be useful.

You might look at the discussion in Burke and Roe (Clim. Dyn., 2014), and Roe and Baker (J. Glac., 2016) for discussion of this in a glaciological context, and at the references therein for more general discussions. There are other references, but I'm most familiar with the ones I've written!

Points:

p1 L4: "This bias could, if not taken into account, imply that the risk of collapse in a given climate change scenario is underestimated." This point is not developed in any way in the paper, and should be removed from the abstract.

p1 L6 approximately 13%. Probably better to say 10 to 15%, given the model simplifications and uncertainties in its general application.

p1 L7: "Many predicted scenarios of the future climate show an increased variability in temperature over much of the Earth." This needs to be supported by citations or

evidence. In most parts of the world observations are consistent with a linear trend acting on the same interannual variability. Unless supported strongly later in the paper, it is not clear if deserves to be in abstract. As of a final reading, there is no further discussion of this in the paper, and it should be removed from the abstract.

p2 L16 "Greenland –, the West Antarctic –, and" weird dashes in my pdf.

p1 L10. This whole introduction should be contracted. The proximity to a tipping point is not a main focus on the paper. The essential point of the paper is a simpler one about the nonlinearity of the mass balance subjected to climate fluctuations, and the effect is quite small. Uncertainty about tipping points is dominated by much larger effects than those postulated here.

p2 L 23 "However, this approach disregards the effects of interannual variability." Perhaps more importantly it assumes the ice sheet is in equilibrium with the control climate (and implicitly the modern climate), which is unlikely to be true for large ice sheets.

p2 L27. "We develop a general theoretical framework for how forcing variability impact the expected response in a model that exhibits a non-linear response." The nonlinearity of total ablation with respect to temperature is implicit in Weertman (1960, 1976) and explicit in, e.g., Roe and Lindzen (2001).

p2 L34 "The results presented here show explicitly how to account for the effect of unresolved temperature variability." Well, it provides one estimate, it is far from a complete accounting and a replacement for its effects.

p3 L4. "That the SMB of an ice sheet model is nonlinear is well known." Statement depends on precise definition of nonlinear. Perhaps better phrased as 'nonlinear with respect to temperature fluctuations'.

p3 L4. It would be nice if the fundamental physical reason were made clear. Although Roe and Lindzen (2001) could have been clearer, both ablation rate and ablation area increase with temperature. The characteristic parabolic shape of ice sheets means

that total ablation rate scales as ~Tˆ3.

p3 L10. Sub annual temperature variability goes back much further. Early PDD formulations recognized the importance of stochastic fluctuations and included daily variability (e.g., Arnold and MacKay,1964; Reeh, 1991; Calov and Greve, 2005).

p3 L12 "broader class of models." broader than what?

p3 L16 "ice sheet initiated from a mountain glacier." On this scale, it is not relevant that it was initiated from a mountain glacier.

p3, L28 "thus all components of the mass budget are uniquely determined by temperature and volume." This simply repeats the preceding clause.

p3, L29 "This is a vast simplification" Not a scientific phrase.

p3, L26 up to section 3: "Before proceeding with the simple model, we investigate the effect of interannual temperature fluctuations by considering the ice sheet as a simple dynamical system." What follows is much fancier than it needs to be. It is a simple point: total ablation is a nonlinear function of temperature, so +ve and -ve fluctuations do not average to zero. That's it. It does not need dressing up with this language, and it is thus less clear than it could be.

p5L11. Well, you've fit the AR(1) to observations, so it had better get the variance right.

p5L25 "The average mass budget of a colder year and a warmer year is less than the mass budget of a year with a temperature corresponding to the average of "cold" and "warm"; to put it another way: the increased SMB of a single anomalously cold year cannot balance the increased melt from an equally anomalously warm year." This and the equation that follows is a fancy way of saying the obvious. It is a shame that the basic geometric reason is not described simply and clearly, here and elsewhere: the ablation rate scales with temperature, the ablation area scales with Tˆ2 because of the typically parabolic shape of ice sheets. So the total ablation scales as ~Tˆ3. An additional nonlinearity arises because the duration of the melt season also depends

on temperature. So of course linear fluctuations do not average to zero. The paper's message would be stronger if a clear, simple physical description were provided.

p7L1 "Fettweis et al. (2013) compare the output of RCMs forced with multiple future climate scenarios and show that the effect of rising temperature on the GrIS SMB is well described by a third degree polynomial" This is consistent with the cubic scaling of Roe and Lindzen (2001), derived from basic ice-sheet geometry.

p8 L8 "(see also supplementing information)." typo.

p8L14 "Combing these numbers we arrive at a warming of 3.0C in the year 2100 relative to the preindustrial when considering the RCP45 scenario. For this value it is seen in Figure 3 that an additional 0.12C should be added to any constant warming term" First, combining, not combing. Second, some context would be useful here. Uncertainty in transient climate response is approximately a factor of 2 (at 2ish sigma). So the effect described here (0.12C) is pretty small in the scheme of things at 2100. A reader should be given a clear message about what the rate limiting uncertainty is for these problems.

p8L25 "The results above highlight that interannual temperature variability cannot be neglected in long term studies involving ice sheet models." Realistically, there are bigger uncertainties that swamp this effect. So these are strong words.

p10 L5 "Our result may be used to explicitly implement the contribution from the temperature fluctuations in the mass balance schemes before bias correcting due to other possible model deficiencies." Hmmmm. How exactly? The effect has been estimated only from one model calculation and only for the Greenland ice sheet. What confidence is there in the numbers so derived? One would need to know the the uncertainties before the correction could be applied even to Greenland, and what confidence is there is applying the effect to Icelandic, Alaskan, or Patagonian ice caps, or to Antarctica? It would be better to have a physical theory rather than to rely on a calibration based on one model and one location.

p10 L10 "The effect is explained by the curvature, or second derivative, of the mass balance as a function of temperature. A negative curvature gives rise to nonlinear effects meaning that the average mass accumulation resulting from a cold year and a warm year in succession is less than the mass accumulation of two consecutive years having the average temperature of the "warm" and "cold" years." This just states what nonlinear means. Again it is a shame not to have a clear physical description of why this is so, since previous studies long ago articulated the fundamental geometric reasons for this behavior.

p10, Line 14: "the results are transferable to other more realistic models" The authors really should be clearer about this. Transfer to what scales, and to models of what? Alpine glaciers, ice capes, other ice sheets? How can it be transferred? The basic point (mass balance is nonlinear) should be considered, but the quantitative application to other systems is highly uncertain and would need specific calibration to each setting.

p10, L24: "This is an interesting special case of an accumulation dominated mass balance" Accumulation variability dominates the mass balance variability of many maritime glaciers. See Medwedeff and Roe (Clim. Dyn., 2017)

p10 L 12: "meaning that the average mass accumulation resulting from a cold year and a warm year in succession is less than the mass accumulation of two consecutive years having the average temperature of the "warm" and "cold" years". Again, this clause just re-explains what nonlinear means. It would be much more satisfying for a reader to have the physical reasons for the nonlinearity explained. Unless the authors have a different answer and an analysis to support it, the leading reason is likely to be that the ablation rate and ablation area both change with temperature. And for approximately parabolic ice sheets, this renders ablation as approximately cubic with temperature.

The use of left arrows in the supplementary to mean "=" is unconventional symbology. in this field, and I think is unnecessarily confusing.

---

## Referee Comment (RC3) · Anonymous Referee #3 · 8 Jun 2017

Review of manuscript "Influence of temperature fluctuations on equilibrium ice sheet volume" by Troels Bøgeholm Mikkelsen, Aslak Grinsted and Peter Ditlevsen [The Cryosphere Discuss. doi:10.5194/tc-2017-47

General comments This paper describes a study of a minimal ice sheet model of axially symmetric ice sheet resting on a bed that slopes linearly downwards from the center. Authors show that there is significant difference in the steady state volume if the model is forced with interannually variable T as compared with constant T. They further show that the effective temperature change resulting from considering fluctuating temperatures is dependent on the warming scenario. This is an interesting modelling study, but it is difficult to see the relevance of this simple model for more complex system, such as the Greenland Ice Sheet, as authors claim in lines 28-30 on page 11. The leap from

this simple model study to conclusion about equilibrium ice volume of the Greenland Ice sheet is not justified in my opinion. The text needs much editing, and I suggest that authors rewrite most of the sections to improve the coherence of the paper. Same terms are called different names throughout the paper, which makes it very confusing to read, V ÌĞ defined with Equation 2 is called mass balance, dV/dt and SMB, I suggest that authors stick to one name. The introduction section lacks coherence, it is not clear where authors are leading to, with their sentences and are jumping back and forth, for example WAIS instability is mentioned in line 20 page 1 and in again in different context in lines 13-15 page 2 and then jumping to threshold for GrIS. The result of the paper is announced in lines 14-16 on page 1, but not in any connection with the surrounding text. I suggest that complete rewriting of the introduction section be made where the current study is put in context, if that is possible (as said above it is not clear to me what the relevance of this modeling exercise is for any of the ice sheets on Earth). The first figure that shows the dependence of the equation of the ELA height with specific balance is introduced but never revisited or used in the study, as the authors (line 10 page 4) "investigate the effect of interannual temperature fluctuations" before proceeding with the simple model. The relationship between (if any) Eq. 1 and Eq. 2 is not explained and the two seem unrelated. Both K and °C are used, select either one.

Specific comments: Page 1 Line 3, the time scale for weather is usually days or weeks, not interannual, it is therefore strange to call the interannual fluctuations "weather generated" Line 4 How is the "risk of collapse" possibly underestimated when the steady state ice sheet is slightly smaller with interannual T fluctuations than constant T? Lines 5-7 This sentence is very confusing and needs clarifications. Line 5, what temperature variability (spatial, temporal, interannual?) What is the relation between "recent ensemble forecasting" and "present day observed value"? How is the effect "adjusted downward", do you mean that the effect is smaller? This sentence needs rewriting. Line 7 use either "predicted" or "scenario" not both Line 9 it is not clear what "further influence" means here, needs clarification Line 11 clarify what "long term forecasting" means (100.000 years, 100 years?) Line 14 as explained above the context of this pre-

sentation of the results is none and this sentence is strangely placed in the introduction. Line 20, something missing after "preindustrial" (time or value?) Lines 20-24 and 1-2 on page 2 strange sentences that need editing and clarification and some coherence

Page 2 Line 4, something is missing or badly placed () for reference, needs editing Line 5 suggest editing, the sentence is not clear ("is associated with" "likely long-term sea level rise" are strange choices - are those modeling results? Line 6 something missing after "interglacial" (period?) suggest to replace "points toward" with "suggest" Line 9-11 suggest editing, "corresponds to an ice free planed" seems strange here Lines 13-14 need editing (what is realistic future scenario?) missing reference for the statement of WAIS committed to collapse. Line 15 suggest to edit "within reach" do you mean "range"? Line 18-19 suggest editing, the context is not clear, as stated above there is lack of coherence in this whole section. Are you suggesting that your modelling study presented in the paper contributes to estimating whether ice sheet is close to a tipping point? Line 23 something missing, interannual variability of what? Line 24 what is "classic" about the study by Pollard et al (1990)? Line 25 suggest to replace "in" with "of" delete one "constant" - what variability? (interannual? Of T?) Line 26 and 27 suggest to replace "expected" with "computed" Line 27-29 the relationship between the simple minimal model of axially symmetric ice sheet resting on a bed that slopes linearly downwards from the centre and the Greenland surface mass balance, so this sentence is strange, needs clarification and editing Line 30 do you mean spatial or temporal variability?

Page 3 Lines 1-3 need editing, how is this related to the other text? Line 4, nonlinear with respect to what? What does "specifically avoid using monthly climatologies in order to include the effect of interannual variability" mean? Some editing is necessary. Line 5 and line 14 what does "climatology" mean here? Temperature and precipitation? Line 6 What kind of errors? Line 9 Bias correction of what? Lines 14-19 needs clarification and editing Lines 23-24, needs editing and missing reference for the "long term ice sheet study"

Page 4 As discussed above the relationship between Eq. 1 and 2 is not explained and there seems to be none here, how is Eq. 1 applied? Line 13, how do you know that the model is sufficient to illuminate the dynamic effect? It is not clear from the text of the paper.

Page 5 There is sometimes (line 1 and line3) swap of the variables in the equation f(T,V) Line 5 what meaning does the expectation value have? Line 20 replace "has" with "have" (variationS) Line 25 "tend to zero" for what condition?

Page 6 Line 10 what is Wt? Lines 12 some editing is necessary, what process? Line 14 missing reference for the value Line 16 how do you determine that 1 year is sufficient? Line 21 suggest to replace "lower" with "smaller"

Page 7 Figure caption missing "simulation" after "constant temperature" and "fluctuating temperature" Line 7 missing "," after specifically Line 8 suggest to replace "this" with "that"

Page 8 Line 1 sentence is not clear, needs editing Line 3 not clear what "this" means here Line 6 "here we take same approach" does not make sense here, are you "comparing the output of RCM..." (line 4)

Page 9 Line 11 red curveS .. show (delete s) Line 12 blue shaded Line 16 something missing after "preindustrial" Line 19-21 this is confusing, if the SMB of Greenland is -234 GT/a and 3°C warming will cause DeltaSMB to be 30 Gt/y, what does it mean for current mass loss? Line 22 suggest to replace "saturate" with "reach a constant value"

Page 10 Figure caption, explain what deltaT and deltaSMB mean in this context Line 5-6 text needs editing, it is not clear what are meant here.

Page 11 line 2 Suggest to add "simulations" after ice sheet model line 11 as explained above replace "mass balance" with , V ÌĞ or dV/dt lines 11-14 need editing, confusing sentence lines 20-30 needs editing, the relationship of the model result to reality is not clear or justified. Technical corrections:

Please also note the supplement to this comment:
http://www.the-cryosphere-discuss.net/tc-2017-47/tc-2017-47-RC3-supplement.pdf

---

## Author Comment (AC1) · 31 Jul 2017

**Response to Interactive Comments**

Troels Bøgeholm Mikkelsen, Aslak Grinsted and Peter D. Ditlevsen

July 31, 2017

**1 Letter from the authors**

Dear editor and reviewers,

First of all we would like thank all of you for your time and careful reading of our manuscript, as evidenced by your thorough and constructive comments.

We have now gone through your comments and answered them each individually; where a change has been made in the new version of the manuscript this is specified.

After reading your comments and re-reading the introduction it was clear that the abstract and introduction needed a substantial revision. This has resulted in the removal of much of the text (as detailed where appropriate below) and a much more focused and hopefully clearer introduction.

Yours sincerely,
Troels Mikkelsen, Aslak Grinsted and Peter Ditlevsen

**2 Comment by X. Fettweis, May 18**

This paper highlights the impact of taking into account the temperature variability in future projections of the GrIS SMB using a simple idealized model. As shown by Fettweis et al. (2013) and well mentioned in this paper, the temperature dependence of melt is not linear while precip increases linearly with temperature. Therefore, forcing a model with a mean climate or a climate resolving the interannual variability will be different. But, don't forget that such effect is explicitly taken into account in all of the simulations forced by GCMs. However, in the idealized models, this effect is often neglected and this paper evaluates the additional mass loss if the interannual variability is taken into account. This (technical) paper is well written, fits well with TC and deserves to be published.

However, we can not forget that the results presented here (13% of additional mass loss) are only valid for

- idealized models.
- for GrIS (where the future changes will be driven by melt increase) and not Antarctica (where the future changes will be driven by precip changes).

**Comment #1:** We agree that the magnitude of the effect is small in comparison to projections for the 21st century. However, the bias will accumulate over time, and can result in metre scale biases in ice sheet volume after a few millenia (as illustrated in figure 2). The effect can therefore not be neglected in longer experiments such as spin-ups tuned to match present day ice sheet volume. This could bias the final volume in paleoclimatic spin-ups (e.g. in ISMIP6-initMIP, and SeaRISE) because the input forcing series may not include realistic high-frequency variability. We disagree that our main result (equation 7) only applies to GrIS and idealized models. You can apply Equation 7 to total mass balance (including discharge) of the Antarctic ice sheet. You can also apply it to individual catchments, mountain glaciers, and even individual grid-cells. The magnitude of the effect will depend on the level of variability, and how non-linear the response is. There is a growing concern that Antarctic mass loss could be very non-linear in the forcing (e.g. DeConto and Pollard). This potential for a large non-linearity tells us that it is important to take forcing variability into account when modelling Antarctica. We feel that this is an important point to highlight, even if we do not quantify it in this paper.

> Therefore, any references to Antarctica in Abstract and Conclusion should be avoid at maximum (e.g. lines 23-27, pg 11) as the Antarctica SMB is, in a 1st order, linearly temperature dependent and the fact that this effect is already explicitly taken into account in all of the more complex/realistic GCMs forced simulations should be more clearly stated into Abstract and Conclusion (e.g. lines 15-19, pg 11).

**Comment #2:** Clausius-Clayperon is non-linear, but it is a fair point that present day Antarctic accumulation is near-linear (6-9 %/K) for temperature deviations of a few degrees. Our equation 7 therefore predicts a small impact of forcing variability on accumulation. However, there is concern that Antarctic ice discharge may have a much more non-linear response (see above). Equation 7 tells us that variability has the potential to be important. We do not quantify it in this paper, but we feel it is important to highlight the potential issue in this paper.

There is nothing to change in abstract as there we have no results or discussion of Antarctica.

**Change #1:** We have added the following to the introduction: "This mechanism is important for the mass balance of present-day Greenland, but less so for present-day Antarctica where mass loss is dominated solid ice discharge (**?**, p. 1170). There is, however, some concern that Antarctic ice discharge and total mass balance may be highly non-linear. The potential for a large non-linear response of Antarctic mass balance is particularly evident in the simulations from **?**."

> Finally, the fact that the interannual variability of temperature could likely be not the dominant missing process (vs positive feedbacks) in idealized models should clearly be mentioned into the conclusion.

**Comment #3:** Equation 7 applies to all models if they do not explicitly allow for forcing variability. It does not only apply to idealized ice sheet models.

**Change #2:** We state that the effect is small compared to other effects after the analysis of the results by **?**.

**3 Comment by G. Roe, May 28**

From the accompanying materials, it is suggested that the paper has undergone several rounds of review already. The editor has already supplied an impressive round of comments to which the authors have responded.

The principle result is one that is important to recognize in all non-linear dynamic models: when models are calibrated to time-mean inputs, there will be bias in the model coefficients because nonlinearities act to rectify variability in the forcing. In my experience this basic point is not as widely appreciated as it should be in glaciology.

**Comment #4:** Thank you. It has been very hard to get this point across. This issue is important in all non-linear models; not just our simple model, and not just GrIS.

The results suggest that the effect is significant enough for ice sheets the size of Greenland and Antarctica, that the issue must be accounted when making future model projections. I think it is worth making the point in the context of ice sheets, and that the result should be published.

I have three main comments and criticisms.

**3.1  1.**

1. I question how important this effect is relative to other uncertainties. While the point is worth making, the size of the effect the authors find is hardly the rate-limiting uncertainty in ice-sheet projections, or in establishing the likelihood of, or proximity to, tipping points. The authors own calculations suggest the effect of variance is the same as changing the mean temperature by 0.12K. This is obviously very small compared to the spread of uncertainty in model projections of future climate change, polar amplification, and the parameterization of ablation. I think a revised manuscript should discuss the results in relation to other uncertainties; and the asserted importance in the abstract and introduction might be dialed down a bit.

**Comment #5:** We agree that 0.12K is not the limiting uncertainty, and we have added text to highlight this (see also our answer to X. Fettweis above). We now compare the temperature adjustment to the uncertainty estimate from **?**, where they estimate a 90% credible interval of 0.8°C – 3.2°C of the GrIS threshold. Our temperature correction is small compared to this range.

**3.2  2.**

2. The physical reason for the nonlinearity should be clearly described. As of now there is almost no explanation, it is presented as a model fact, and only recent papers are cited. A reader will likely crave having a physical reason provided.

**Comment #6:** We have rewritten the introduction to better align with the focus of the paper. In this new introduction we explain the non-linearity. Further, (as you suggest below) we now cite **?** several places where appropriate.

**Change #3:** The following has been added to the introduction: "Ice sheets are characterized by a large interior plateau flanked by comparatively steeper margins. A warming will shift the equilibrium line altitude (ELA) to higher elevations increasing the area exposed to melt. The area exposed to melt will increase non-linearly with ELA because of the top-heavy hypsometry (**?**)."

> That the mass balance should be nonlinear has long been known. It is implicit in the ELA sensitivities derived sixty years ago in Wertman (J Glac, 1960, 1963, Science 1976). The reason is also explicitly derived in Roe and Lindzen (Clim Dyn, 2001), and likely earlier and elsewhere. The ablation rate scales as temperature, and the ablation area scale as $\sim T^2$ because of the characteristic parabolic profile of ice sheets; giving a roughly cubic dependency for total ablation. There is also a smaller, but nontrivial effect, that the length of the melt season changes with T. Note the degree of nonlinearity will be different for the plastic-rheology profiles the authors use in the Oer03, from the dynamic ice model used in Robinson et al., so there is an internal inconsistency in the results presented here. (The degree of nonlinearity is larger for a shallow-ice rheology than it is for plastic-rheology ice sheet, by an increase of approximately one in the exponent.)

**Comment #7:** Thank you for the pointers to existing literature. We were aware of these, but did not find them necessary originally.

Clarification: The calculations we do on **?** are done using the curvature we measure from Robinssons results. There is no inconsistency.

> I don't understand why the authors did not use the temperature nonlinearity that is directly represented in the Oer03 model, and instead calibrated to a completely different model set-up from Robinson et al. (the former is an axisymmetric ice sheet and climate, the latter is a realistic Greenland). In Oer03 the ELA is directly specified in terms of temperature, and the geometric nonlinearity in the ablation-temperature relation certainly exists in Oer03. It would be a more self-consistent estimate of the effect, and certainly worth comparing with the extrapolations from Robinson et al.

**Comment #8:** The Oer03 model is only illustrative, and not useful in a quantitative sense. So, we cannot apply a correction based on the second derivative of Oer03 to Robinsson.

Here's our thinking behind the order of presentation in the paper: We derive equation 7, and we want to explain it. This equation applies to all models, and so we choose to use a simple model to illustrate how it works. Perfectly plastic models such as Oer03 are attractive here because the mass balance can be written as a function of $V$ and $T$ alone. This enables us to make the Fig. 2 which we think is useful. We do not argue that Oer03 is realistic, and we only use it for illustrative purposes. We use it to verify that equation 7 holds for this simple case.

Once we have shown the reader that it works on the simple model, then we proceed to

- highlight that the non-linear relationship between T and mass-balance is not just an artefact of the simple Oer03 model. It is also present in RCM results from Fettweis.

- Estimate the magnitude of the effect on the results from a real ice sheet model **?**.

If the authors have other mechanisms they have diagnosed or have speculations about, those should be given too. Otherwise it can be frustrating to read about an effect whose cause is not explained.

**Comment #9:** If this comment refers to the nonlinear relationship between temperature and SMB, we think manuscript has been improved by the edited introduction (see Change #3) and added references to **?**.

**3.3  3.**

3. The application of stochastic climate variability.

The authors represent stochastic variability by applying AR(1) red noise in annual-mean temperatures.

Applying stochastic variability to the annual mean temperature is likely wrong. Annual-mean anomalies are the result of much larger stochastic variation in seasonal temperatures (seasonal fluctuations are $\sim$ sqrt(4) larger than annual mean. A model will fail to emulate realistic mass balance anomalies without accounting for theses larger seasonal fluctuations that actually drive the ablation budget.

**Comment #10:** It is clear that we need to use the full variance in equation 6. However, sub-annual frequency variability may already have been accounted for in the model. E.g. **?** already model the seasonal cycle, and the Oer03 PDD factors have been selected to be representative over an entire year. Our interpretation of these models mean we only need to consider variability on annual and longer timescales.

p5L9 "(AR2 AR10 ) = (0.67, 0.85)" What is the persistence timescale implied by this coefficient? For AR(1) tau = 1yr/(1-a) giving tau $\sim$ 3yrs. The uninitiated reader has no idea what the point of AR(1) is, and why it is important to use, so more explanation is needed. The persistence in the annual-mean anomalies are not going to be the same as the melt-season anomalies, which is important to account for. The ablation anomaly is due to the melt-season temperatures, and the melt-season persistence timescale is typically less than annual-mean.

Furthermore, for this to be rigorous, some kind of criterion should be used to evaluate whether AR(1) is a sufficient, self-consistent, and parsimonious description of the data.

There are various methods for establishing this (e.g.., Akaike information criteria, etc.), but none are referred to, so a reader has no idea of the necessity of this fit, or if any such estimate was performed. The data should be detrended before fitting, and the residuals should be tested for any remaining autocorrelation. I did not find either of these mentioned in the paper or supplementary.

**Comment #11:** Visually it is clear that there is some autocorrelation. We use a minimal model of that, but it is really not that important. Equation 6 does not depend on the autocovariance structure. Hence, we feel that it would be a distraction to discuss alternative models. One could score the models using Akaike information criteria, or similar to justify any more complex model.

We have done analyses similar to the ones presented in the paper (these additional analyses are not shown) using temperature time series $T_t$ consisting of either 1) Gaussian random variates (white noise) and AR(1)-generated noise with a much longer persistence time scale $\tau$ (i.e., *not* a fit to observed Greenland temperatures). The persistence time scale of $T_t$ naturally affects the persistence of $V_t$, but the change in equilibrium volume can still be determined using equation 6 in the manuscript and the *variance* of $T_t$ alone, without referring to the persistence time scale.

We feel that an in-depth discussion of these issues would distract from our main point and add comparatively little to the quality of the paper.

Just by eye, the temperature time series shown in the supplementary looks a bit questionable between 1850 and 1900. I would recommend subsampling the data, and using other datasets (preferably instrumental records) to see how stable the estimates are.

**Comment #12:** As per above, correctly modeling the observed mean temperature over Greenland is not our main goal in the article. Such a study done properly would probably make a pretty good manuscript on its own.

Finally, AR(1) is a somewhat limited representation of climatic persistence. The spectrum flattens out at periods longer than 2*pi*tau, and so there is no persistence at multidecadal and centennial scales, in contrast to a power-law representation, for instance. The nature of climatic persistence at low frequencies is debated, but alternative representations would have important implications for these results and affect the answers quantitatively, so some discussion would be useful.

You might look at the discussion in Burke and Roe (Clim. Dyn., 2014), and Roe and Baker (J. Glac., 2016) for discussion of this in a glaciological context, and at the references therein for more general discussions. There are other references, but I'm most familiar with the ones I've written!

**Comment #13:** Long range persistence vs. AR(1) type models is an interesting issue and would probably warrant yet another paper in itself. Equation 6 is only sensitive to the variance; we allow for the simplest possible model of persistence.

**3.4 Points:**

p1 L4: "This bias could, if not taken into account, imply that the risk of collapse in a given climate change scenario is underestimated." This point is not developed in any way in the paper, and should be removed from the abstract.

**Comment #14:** AR5 put the collapse threshold for GrIS at +1.6 °C. The source for that is ultimately Robinsson et al. We estimate that the Robinsson results has to be shifted by -0.12°C. Do we really need to develop it further than that in the paper to justify that sentence?

It certainly seems important enough to highlight in the abstract, when you hold it up against the targets that have been put forward politically.

p1 L6 approximately 13%. Probably better to say 10 to 15%, given the model simplifications and uncertainties in its general application.

**Comment #15:** We agree that credible intervals for the $\Delta T$ and $\Delta$SMB estimates should be added to the text. In the previous version of the manuscript this could only be determined from Fig. 4.

**Change #4:** The abstract now reads: "We estimate the bias to be 30 Gt/y (24 Gt/y – 59 Gt/y, 95% credibility) for a warming of 3 °C above preindustrial values, or 13% (10% – 25%, 95% credibility) of the present day rate of ice loss."

p1 L7: "Many predicted scenarios of the future climate show an increased variability in temperature over much of the Earth." This needs to be supported by citations or evidence. In most parts of the world observations are consistent with a linear trend acting on the same interannual variability. Unless supported strongly later in the paper, it is not clear if deserves to be in abstract. As of a final reading, there is no further discussion of this in the paper, and it should be removed from the abstract.

**Change #5:** We agree – this sentence has been removed.

p2 L16 "Greenland –, the West Antarctic –, and" weird dashes in my pdf.

**Change #6:** Sentence has been removed.

p1 L10. This whole introduction should be contracted. The proximity to a tipping point is not a main focus on the paper. The essential point of the paper is a simpler one about the nonlinearity of the mass balance subjected to climate fluctuations, and the effect is quite small. Uncertainty about tipping points is dominated by much larger effects than those postulated here.

**Comment #16:** It is correct that the main point of the paper is about the nonlinearity of the mass balance subjected to climate fluctuations. However, the mass balance non-linearity is more pronounced as you get closer to the threshold.

AR5 show the Greenland tipping point in figure 13.14c based on the model from **?**. We estimate that this might be biased by -0.12°C. AR5 show a 90%

credible interval from 0.8 to 2.2°C for the threshold location. This corresponds to a standard deviation of 0.42°C, assuming normality. So, yes, you are right that the full uncertainty is greater than the effect we discuss, but it is hardly insignificant. And further the effect we discuss is a bias with a clear sign.

> p2 L 23 "However, this approach disregards the effects of interannual variability." Perhaps more importantly it assumes the ice sheet is in equilibrium with the control climate (and implicitly the modern climate), which is unlikely to be true for large ice sheets.

**Comment #17:** This is a very good point. While editing the introduction we have striven to make it as compact and direct as possible; we feel that mentioning this point would add too much length.

> p2 L27. "We develop a general theoretical framework for how forcing variability impact the expected response in a model that exhibits a non-linear response." The nonlinearity of total ablation with respect to temperature is implicit in Weertman (1960, 1976) and explicit in, e.g., Roe and Lindzen (2001).

**Comment #18:** Thank you for this point – we have added these references to the introduction.    **Change #7:** The introduction now opens with "Ice sheet mass balance has a non-linear relationship with temperature. This behavior is observed in simple ice sheet models (????), in full regional climate modeling of Greenland surface mass balance (?), and the non-linear effect between temperature and melt has been observed in Greenland river discharge (?)."

> p2 L34 "The results presented here show explicitly how to account for the effect of unresolved temperature variability." Well, it provides one estimate, it is far from a complete accounting and a replacement for its effects.

**Comment #19:** We agree, that sentence is incorrect.    **Change #8:** We have rephrased to "Our contribution is a quantification of this effect, and an estimate of the necessary bias correction needed to account for temperature fluctuations in long term ice sheet simulations."

> p3 L4. "That the SMB of an ice sheet model is nonlinear is well known." Statement depends on precise definition of nonlinear. Perhaps better phrased as 'nonlinear with respect to temperature fluctuations'.

**Change #9:** We have rephrased to "nonlinear with respect to temperature".

> p3 L4. It would be nice if the fundamental physical reason were made clear. Although Roe and Lindzen (2001) could have been clearer, both ablation rate and ablation area increase with temperature. The characteristic parabolic shape of ice sheets means that total ablation rate scales as $\sim T^3$.

**Change #10:** We have added a the following after the sentence at p3 L4: "In a simplified model of continental ice sheets, **?** show that the total annual ablation scales with the cube of temperature at the ice sheet margin."

p3 L10. Sub annual temperature variability goes back much further. Early PDD formulations recognized the importance of stochastic fluctuations and included daily variability (e.g., Arnold and MacKay,1964; Reeh, 1991; Calov and Greve, 2005).

**Change #11:** Thank you for this point – we have added these references to the introduction.

p3 L12 "broader class of models." broader than what?

**Change #12:** We have removed that sentence.

p3 L16 "ice sheet initiated from a mountain glacier." On this scale, it is not relevant that it was initiated from a mountain glacier.

**Change #13:** Agree – we have removed that sentence.

p3, L28 "thus all components of the mass budget are uniquely determined by temperature and volume." This simply repeats the preceding clause.

**Change #14:** Good point, we have removed that sentence.

p3, L29 "This is a vast simplification" Not a scientific phrase.

**Change #15:** We have removed that sentence.

p3, L26 up to section 3: "Before proceeding with the simple model, we investigate the effect of interannual temperature fluctuations by considering the ice sheet as a simple dynamical system." What follows is much fancier than it needs to be. It is a simple point: total ablation is a nonlinear function of temperature, so +ve and -ve fluctuations do not average to zero. That's it. It does not need dressing up with this language, and it is thus less clear than it could be.

**Comment #20:** We do not agree that section 2.1 from "simple dynamical system" and onwards is written in an overly complicated way. It is clear that 'hot' and 'cold' do not average to zero, and that this is relatively straightforward to explain. The main contribution of this work, however, is equation 6 and to derive that we need the techniques from dynamical systems theory.

p5L11. Well, you've fit the AR(1) to observations, so it had better get the variance right.

**Change #16:** We agree that this sentence is superfluous – it has been removed.

p5L25 "The average mass budget of a colder year and a warmer year is less than the mass budget of a year with a temperature corresponding to the average of "cold" and "warm"; to put it another way: the increased SMB of a single anomalously cold year cannot balance the increased melt from an equally anomalously warm year." This and

the equation that follows is a fancy way of saying the obvious. It is a shame that the basic geometric reason is not described simply and clearly, here and elsewhere: the ablation rate scales with temperature, the ablation area scales with $T^2$ because of the typically parabolic shape of ice sheets. So the total ablation scales as $\sim T^3$. An additional nonlinearity arises because the duration of the melt season also depends on temperature. So of course linear fluctuations do not average to zero. The paper's message would be stronger if a clear, simple physical description were provided.

**Comment #21:** Thank you for this comment – we have added a reference to **?**, as stated above.

p7L1 "Fettweis et al. (2013) compare the output of RCMs forced with multiple future climate scenarios and show that the effect of rising temperature on the GrIS SMB is well described by a third degree polynomial" This is consistent with the cubic scaling of Roe and Lindzen (2001), derived from basic ice-sheet geometry.

**Change #17:** We have added the following: "consistent with the aforementioned findings of **?**."

p8 L8 "(see also supplementing information)." typo.

**Change #18:** Fixed.

p8L14 "Combing these numbers we arrive at a warming of 3.0C in the year 2100 relative to the preindustrial when considering the RCP45 scenario. For this value it is seen in Figure 3 that an additional 0.12C should be added to any constant warming term" First, combining, not combing. Second, some context would be useful here. Uncertainty in transient climate response is approximately a factor of 2 (at 2ish sigma). So the effect described here (0.12C) is pretty small in the scheme of things at 2100. A reader should be given a clear message about what the rate limiting uncertainty is for these problems.

p8L25 "The results above highlight that interannual temperature variability cannot be neglected in long term studies involving ice sheet models." Realistically, there are bigger uncertainties that swamp this effect. So these are strong words.

**Comment #22:** As stated above, the effect is small but nevertheless a bias with a clear sign.

**Change #19:** We have added a 95% credible interval to the estimate.

**Change #20:** We believe we now spell "combining" correctly!

p10 L5 "Our result may be used to explicitly implement the contribution from the temperature fluctuations in the mass balance schemes before bias correcting due to other possible model deficiencies." Hmmmm. How exactly? The effect has been estimated only from one model calculation and only for the Greenland ice sheet. What confidence is there in the numbers so derived? One would need to know

the the uncertainties before the correction could be applied even to Greenland, and what confidence is there is applying the effect to Icelandic, Alaskan, or Patagonian ice caps, or to Antarctica? It would be better to have a physical theory rather than to rely on a calibration based on one model and one location.

**Comment #23:** We agree that this wording is too strong. On the other hand we have shown that in a simple model this effect cannot be neglected, hopefully prompting further investigations using more comprehensive models.

**Change #21:** We have rephrased to "Our results shows the importance of considering temperature fluctuations in the mass balance schemes before bias correcting for other possible model deficiencies."

p10 L10 "The effect is explained by the curvature, or second derivative, of the mass balance as a function of temperature. A negative curvature gives rise to nonlinear effects meaning that the average mass accumulation resulting from a cold year and a warm year in succession is less than the mass accumulation of two consecutive years having the average temperature of the "warm" and "cold" years." This just states what nonlinear means. Again it is a shame not to have a clear physical description of why this is so, since previous studies long ago articulated the fundamental geometric reasons for this behavior.

**Comment #24:** We agree that the two sentences basically state the same, but added the second sentence as a potentially useful clarification for the reader. Regarding the physical explanation, the reader may refer to the previously added references to **?**.     **Change #22:** We have removed the sentence starting with "A negative curvature . . . "

p10, Line 14: "the results are transferable to other more realistic models" The authors really should be clearer about this. Transfer to what scales, and to models of what? Alpine glaciers, ice capes, other ice sheets? How can it be transferred? The basic point (mass balance is nonlinear) should be considered, but the quantitative application to other systems is highly uncertain and would need specific calibration to each setting.

**Comment #25:** We agree that this sentence could be clearer.

**Change #23:** We have rephrased the sentence to: "We find that the steady state ice sheet volume in Oer03 is $0.5 - 1$ $\mathrm{m_{SLE}}$ smaller when the minimal model is forced with fluctuating temperatures compared to constant temperature (Fig. 2). It is therefore necessary to consider the impact of temperature variability when designing long-term model experiments such as paleo spin-ups (eg. **???**), especially when downsampling the paleo forcing series. Furthermore, models of sub-shelf melting, grounding line migration, and ice discharge have the potential to respond non-linearly to changes in ocean temperatures (**??????**), thus it is critical to take variability into account for quantitative assessments."

p10, L24: "This is an interesting special case of an accumulation dominated mass balance" Accumulation variability dominates the mass balance variability of many maritime glaciers. See Medwedeff and Roe (Clim. Dyn., 2017)

**Comment #26:** This is a very interesting reference, thank you. However, we feel that the specific connection between temperature and precipitation regarding Antarctica is best illustrated by **?**.

> p10 L 12: "meaning that the average mass accumulation resulting from a cold year and a warm year in succession is less than the mass accumulation of two consecutive years having the average temperature of the "warm" and "cold" years". Again, this clause just re-explains what nonlinear means. It would be much more satisfying for a reader to have the physical reasons for the nonlinearity explained. Unless the authors have a different answer and an analysis to support it, the leading reason is likely to be that the ablation rate and ablation area both change with temperature. And for approximately parabolic ice sheets, this renders ablation as approximately cubic with temperature.

**Change #24:** Thank you – we have removed that sentence.

> The use of left arrows in the supplementary to mean "=" is unconventional symbology. in this field, and I think is unnecessarily confusing.

**Comment #27:** This is a valid point, but we do think that left arrows for assignment helps distinguish between an algorithm (such as the Oer03 model) and the derivation of a result.

**4   Comment by Anonymous Referee #3, June 8**

> Review of manuscript "Influence of temperature fluctuations on equilibrium ice sheet volume" by Troels Bøgeholm Mikkelsen, Aslak Grinsted and Peter Ditlevsen [The Cryosphere Discuss. doi:10.5194/tc-2017-47
>
> **General comments:** This paper describes a study of a minimal ice sheet model of axially symmetric ice sheet resting on a bed that slopes linearly downwards from the center. Authors show that there is significant difference in the steady state volume if the model is forced with interannually variable T as compared with constant T. They further show that the effective temperature change resulting from considering fluctuating temperatures is dependent on the warming scenario. This is an interesting modelling study, but it is difficult to see the relevance of this simple model for more complex system, such as the Greenland Ice Sheet, as authors claim in lines 28-30 on page 11.
>
> The leap from this simple model study to conclusion about equilibrium ice volume of the Greenland Ice sheet is not justified in my opinion.

**Comment #28:** It is likely that we did not in a sufficiently direct way explain that the output of Oer03 is *not* the main result of this paper. The study

is not about the minimal ice sheet model, which is purely illustrative. The main result of the paper is equation 6, and the implications it has.

Equation 6 is valid for all models regardless of complexity. Whether it is a significant effect depend on

1. the variance of the forcing.

2. how non-linear the response is.

We first use Oer03 to illustrate how equation 6 works (and verify that it is correct). We then proceed to show that the mass-balance nonlinearity is also present in RCM results from **?**. **?** estimate a collapse threshold of Greenland at around $+1.6°C$. Using equation 7 we find that this needs to be adjusted by about $0 - -0.2°C$. This is not a huge bias considering all other uncertainties, but it is hardly insignificant considering how close we are to that threshold already.

Using Oer03 we show that the bias can result in metre scale volume differences. This can be important for 'real' ice sheet models in paleo spin-up simulations such as those from SeaRISE and initMIP/ISMIP6. The forcing used in the paleo spin up experiments is unlikely to have a realistic representation of the variance; e.g. the CISM SeaRISE setup for Greenland uses a forcing series with no data between 6400 and 0 BP, and 100 year resolution prior to that (`http://websrv.cs.umt.edu/isis/index.php/Configuration\_Files`). Similarly our results highlight that you will introduce a bias if you force the model with ensemble mean/median. This is apparently what was done for the CISM SeaRISE experiments (`http://websrv.cs.umt.edu/isis/index.php/Future\_Climate\_Data`).

> The text needs much editing, and I suggest that authors rewrite most of the sections to improve the coherence of the paper. Same terms are called different names throughout the paper, which makes it very confusing to read, $\dot{V}$ defined with Equation 2 is called mass balance, dV/dt and SMB, I suggest that authors stick to one name.

**Change #25:** Agree – we have changed all occurrences of $\dot{V}$ to $dV/dt$.

> The introduction section lacks coherence, it is not clear where authors are leading to, with their sentences and are jumping back and forth, for example WAIS instability is mentioned in line 20 page 1 and in again in different context in lines 13-15 page 2 and then jumping to threshold for GrIS.

**Comment #29:** We agree and have rewritten the intro with a stronger focus.

> The result of the paper is announced in lines 14-16 on page 1, but not in any connection with the surrounding text. I suggest that complete rewriting of the introduction section be made where the current study is put in context, if that is possible (as said above it is not clear to me what the relevance of this modeling exercise is for any of the ice sheets on Earth).

**Comment #30:** We agree that this sentence is out of place – we have rewritten the intro.

The first figure that shows the dependence of the equation of the ELA height with specific balance is introduced but never revisited or used in the study, as the authors (line 10 page 4) "investigate the effect of interannual temperature fluctuations" before proceeding with the simple model.

**Comment #31:** In this section we aim to make a minimal description of the Oer03 model without going into any detail. Fig. 1 was moved from the supplementing information to the main manuscript due to a request from the editor.

The relationship between (if any) Eq. 1 and Eq. 2 is not explained and the two seem unrelated.

**Comment #32:** We have chosen to place the details of the Oer03 model in the supplementing information, where the relationship between equations 1 and 2 are specified. We agree that this was not clear from the manuscript.

**Change #26:** We specify in the manuscript that details – including the explicit connection between equations 1 and 2 – are found in the supplementing information.

Both K and °C are used, select either one.

**Change #27:** We have chosen °C as the temperature unit.

**Specific comments – Page 1**

Line 3, the time scale for weather is usually days or weeks, not interannual, it is therefore strange to call the interannual fluctuations "weather generated"

**Change #28:** We have changed the wording from "interannual weather generated fluctuations" to "interannual fluctuations".

Line 4 How is the "risk of collapse" possibly underestimated when the steady state ice sheet is slightly smaller with interannual T fluctuations than constant T?

**Comment #33:** With no fluctuations the ice sheet is smaller (as you correctly say). A smaller ice sheet is closer to the threshold. **?** estimate the threshold at $0.8 - -3.2$°C. We calculate that if you apply our bias correction then this should be shifted by $-0.2 - -0.0$°C. I.e. we are closer to the threshold.

Lines 5-7 This sentence is very confusing and needs clarifications.

**Comment #34:** We agree and have rewritten the abstract.

Line 5, what temperature variability (spatial, temporal, interannual?) What is the relation between "recent ensemble forecasting" and "present day observed value"? How is the effect "adjusted downward", do you mean that the effect is smaller? This sentence needs rewriting.

**Comment #35:** We agree and have rewritten the abstract.

Line 7 use either "predicted" or "scenario" not both

**Change #29:** We will stick with "scenario".

Line 9 it is not clear what "further influence" means here, needs clarification

**Change #30:** We have removed the word "further"

Line 11 clarify what "long term forecasting" means (100.000 years, 100 years?)

**Comment #36:** When we use the phrase "long term" to refer to the study by **?** we mean "on a time scale were a complete or substantial melt of the GrIS is feasible". Depending on the warming this will of course vary (see eg. Fig. 3b in **?**). With this in mind, we believe that the phrase "long term" is justified, even without stating a specific number of years.

Line 14 as explained above the context of this presentation of the results is none and this sentence is strangely placed in the introduction.

**Comment #37:** Agree - the intro has been rewritten with a stronger focus.

Line 20, something missing after "preindustrial" (time or value?)

**Change #31:** We have added "value".

Lines 20-24 and 1-2 on page 2 strange sentences that need editing and clarification and some coherence

**Change #32:** We have removed this sentence when editing the introduction.

**Page 2**

Line 4, something is missing or badly placed () for reference, needs editing

**Change #33:** We removed this sentence during the editing of the introduction.

Line 5 suggest editing, the sentence is not clear ("is associated with" "likely long-term sea level rise" are strange choices - are those modeling results?

**Change #34:** That sentence has been removed.

Line 6 something missing after "interglacial" (period?) suggest to replace "points toward" with "suggest"

**Change #35:** Sentence removed.

Line 9-11 suggest editing, "corresponds to an ice free planed" seems strange here

**Change #36:** Sentence removed.

Lines 13-14 need editing (what is realistic future scenario?) missing reference for the statement of WAIS committed to collapse.

**Change #37:** Sentence removed.

Line 15 suggest to edit "within reach" do you mean "range"?

**Change #38:** Sentence removed.

Line 18-19 suggest editing, the context is not clear, as stated above there is lack of coherence in this whole section. Are you suggesting that your modelling study presented in the paper contributes to estimating whether ice sheet is close to a tipping point?

**Change #39:** Sentence removed.

Line 23 something missing, interannual variability of what?

**Change #40:** Changed to "interannual temperature variability".

Line 24 what is "classic" about the study by Pollard et al (1990)?

**Change #41:** Sentence removed.

Line 25 suggest to replace "in" with "of" delete one "constant" - what variability? (interannual? Of T?)

**Change #42:** Sentence removed.

Line 26 and 27 suggest to replace "expected" with "computed"

**Change #43:** Sentence removed.

Line 27-29 the relationship between the simple minimal model of axially symmetric ice sheet resting on a bed that slopes linearly downwards from the centre and the Greenland surface mass balance, so this sentence is strange, needs clarification and editing

**Comment #38:** We have rewritten the introduction so that this sentence no longer appears. It is however clear that the Oer03 is an idealized model. To model the future of the GrIS with Oer03 would indeed be futile. We use the Oer03 as it has the practical property for the derivation of equation 6 that $dV/dt = f(T, V)$. Further, $dV/dt$ in Oer03 has a qualitatively similar relationship to temperature (Fig. 3, left) as the model used in **?** (Fig. 3, right) and the relationship found by **?** (Fig. 6h in that article) so we believe the use of Oer03 for the present purpose is justified.

Line 30 do you mean spatial or temporal variability?

**Change #44:** We meant temporal variability – the sentence has been removed.

**Page 3**

Lines 1-3 need editing, how is this related to the other text?

**Comment #39:** The mentioning of **?** and **?** were requested by a previous reviewer. As these studies concern the variability of GrIS SMB under a warming climate (**?**) or the influence of fast temperature fluctuations on glacier extent (**?**), we believe they provide valuable context for the reader.

Line 4, nonlinear with respect to what?

**Change #45:** With respect to temperature – this information has been added.

What does "specifically avoid using monthly climatologies in order to include the effect of interannual variability" mean? Some editing is necessary.

**Change #46:** We have changed the wording to "**?** specifically avoid using average monthly temperature and precipitation climatologies and instead use time series from individual months in order to include the effect of interannual variability in their study."

Line 5 and line 14 what does "climatology" mean here? Temperature and precipitation?

**Comment #40:** Yes, air temperature and precipitation in both cases.
   **Change #47:** This information has been added.

Line 6 What kind of errors?

**Change #48:** We have added "... errors, either over- or underestimating the the SMB relative to a computed reference SMB."

Line 9 Bias correction of what?

**Change #49:** We have changed the wording to "... necessary bias correction needed to account for temperature fluctuations in long term ice sheet simulations."

Lines 14-19 needs clarification and editing

**Change #50:** We have specified that the different "formulations" used by **?** refer to the simplifying assumptions employed in the PDD formulation.

Lines 23-24, needs editing and missing reference for the "long term ice sheet study"

**Change #51:** We have added a reference to **?** and specify that we investigate the effect of temperature fluctuations on their study.

**Page 4**

> As discussed above the relationship between Eq. 1 and 2 is not explained and there seems to be none here, how is Eq. 1 applied?

**Comment #41:** We have chosen to place a detailed description of the model in the supplementing information, in order to not distract the reader from our main point. In the supplement, equation 1 from the manuscript concerning $h_{Eq}$(the altitude of the equilibrium line) is shown as equation 2; we then give a detailed description of how $h_{Eq}$ enters the calculations.
  **Change #52:** We have added a reference to the supplementing information between Equations 1 and 2.

> Line 13, how do you know that the model is sufficient to illuminate the dynamic effect? It is not clear from the text of the paper.

**Comment #42:** We take the qualitative similarity of the $SMB(T)$ curves of the Oer03, the results by **?** and the results by **?** as an indicator that our method is justified. We do not attempt to model the GrIS with the Oer03, we use it to illustrate the effect of temperature variability.

**Page 5**

> There is sometimes (line 1 and line3) swap of the variables in the equation f(T,V)

**Change #53:** We believe we have caught and eliminated all use of $f(V,T)$ so that only $f(T,V)$ remains.

> Line 5 what meaning does the expectation value have?

**Comment #43:** The expectation value of the temperature $\overline{T} = \langle T_t \rangle$ is the time average of a fluctuating temperature time series, in this manuscript generated with equation 8. This is varied in the simulations shown in Fig. 2 where $T_t$ can be read on the $x$-axis in the right panel. $\overline{V} = \langle V_t \rangle$ is the time average of ice sheet volume when the model has reached steady state. This corresponds to the $y$-axis in Fig. 2 (both panels).
  **Change #54:** We have added the following before the derivation equation 6: "In Fig. 2, $\overline{T}$ is shown on the horizontal axis in the right panel, and the corresponding $\overline{V}$ on the vertical axis (both panels)."

> Line 20 replace "has" with "have" (variationS)

**Change #55:** Fixed.

> Line 25 "tend to zero" for what condition?

**Comment #44:** Thank you, this was unclear in the manuscript.
  **Change #56:** We have added "tend to zero *as the ice sheet approaches equilibrium volume*".

**Page 6**

Line 10 what is Wt?

**Change #57:** We have added "where $W_t, t = 1, 2, \ldots$ are independent, random draws from a standard normal distribution."

Lines 12 some editing is necessary, what process?

**Change #58:** Thank you – we have clarified that we are referring to the *autoregressive* process defined by equation 8.

Line 14 missing reference for the value

**Comment #45:** This sentence has been removed in response to G. Roe's comment above. For reference, the value is the observed variance of the red curve in Fig. 2 (supplementing information).

Line 16 how do you determine that 1 year is sufficient?

**Comment #46:** We determine that a step size $\Delta t$ used for integration of the Oer03 model to be sufficient by using the same simulated temperature time series for varying $\Delta t$ (Fig. 1, supplementing information). We use $\Delta t$ values of 1, 0.5, 0.1 and 0.01 years, respectively. In this way the same temperature is used for the whole year in the Oer03 model, and the model integrated with 1, 2, 10 or 100 time steps per year. As the results are practically identical based on a visual inspection of the resulting ice volume curves, we conclude that $\Delta t = 1$ year is sufficient. Please note the scale on the $V$-axis in Fig. 1 (supplement) compared to the Fig. 2 (article).
   **Change #59:** We have added a reference in the manuscript specifically to *Fig. 1* in the supplement.

Line 21 suggest to replace "lower" with "smaller"

**Change #60:** We have substituted "smaller".

**Page 7**

Figure caption missing "simulation" after "constant temperature" and "fluctuating temperature"

**Change #61:** We have changed the wording from "" to ""

Line 7 missing "," after specifically

**Change #62:** Fixed.

Line 8 suggest to replace "this" with "that"

**Change #63:** We agree and have made the change.

**Page 8**

Line 1 sentence is not clear, needs editing

**Change #64:** We have changed the wording to "As parameters in ice sheet models are often tuned to reproduce an observed ice sheet history from a time series of forcing observations (eg., **?**), ..."

Line 3 not clear what "this" means here

**Change #65:** Added "bias"

Line 6 "here we take same approach" does not make sense here, are you "com- paring the output of RCM. . ." (line 4)

**Comment #47:** We are referring to fitting third degree polynomials, not comparing different RCMs – this was unclear, thank you.
  **Change #66:** Change to "We will follow **?** and to the ensemble of simulations in **?** fit third degree polynomials ..."

**Page 9**

Line 11 red curveS .. show (delete s)

**Change #67:** Fixed, thank you.

Line 12 blue shaded

**Change #68:** Typo fixed, thank you.

Line 16 something missing after "preindustrial"

**Change #69:** Added "period" after "preindustrial".

Line 19-21 this is confusing, if the SMB of Greenland is -234 GT/a and 3°C warming will cause DeltaSMB to be 30 Gt/y, what does it mean for current mass loss?

**Comment #48:** We provide the SMB of Greenland to put the magnitude of the effect we describe into context. What temperature fluctuations mean for the current SMB is a very interesting question. We do not presume to answer this question in the present manuscript; to determine this would likely require model studies employing more sophisticated ice sheet models than the Oer03 used here, and we hope that such studies will be carried out in the future. We agree that the phrasing was unclear and have changed it accordingly.
  **Change #70:** We have changed the wording from to "Fig. 4 (bottom) shows the most likely $\Delta$SMB resulting from temperature fluctuations at a 3°C warming to be 30 Gt/y or – for context – 12.8% of the average GrIS SMB of $-234 \pm 20$ Gt/y reported for the period 2003–2011 (**?**)."

Line 22 suggest to replace "saturate" with "reach a constant value"

**Change #71:** This change has been made.

**Page 10**

Figure caption, explain what deltaT and deltaSMB mean in this context

**Change #72:** We have added this description here.

Line 5-6 text needs editing, it is not clear what are meant here.

**Change #73:** We have edited to "Temperature fluctuations can be accounted for in ice sheet modeling studies, either explicitly (eg. **??**) or implicitly, as happens when tuning the ice sheet model to reproduce an observed ice sheet history with observed forcing as input."

**Page 11**

Page 11 line 2 Suggest to add "simulations" after ice sheet model

**Change #74:** We agree and have changed the text accordingly.

line 11 as explained above replace "mass balance" with , $\dot{V}$ or dV/dt

**Comment #49:** The use of "mass balance", $\dot{V}$ etc. has been cleaned up substantially as per your previous request – in this particular place we think it is better to be verbose.

lines 11-14 need editing, confusing sentence

**Change #75:** We have changed the wording to "A negative curvature gives rise to nonlinear effects where the mass balance anomaly of a cold year is insufficient to offset the mass balance anomaly from a warm year of equal magnitude"

lines 20-30 needs editing, the relationship of the model result to reality is not clear or justified.

**Comment #50:** As previously stated, the qualitative similarity of the SMB($T$) curves in Oer03, **?** and **?** justifies our approach; the SMB $\sim T^3$ relationship is further supported by **?**. However, the conclusion has been rewritten, as detailed above in the answer to G. Roe; we repeat the relevant part for convenience: "We find that the steady state ice sheet volume in Oer03 is $0.5 - 1$ m$_{SLE}$ smaller when the minimal model is forced with fluctuating temperatures compared to constant temperature (Fig. 2). It is therefore necessary to consider the impact of temperature variability when designing long-term model experiments such as paleo spin-ups (eg. **???**), especially when downsampling the paleo forcing series. Furthermore, models of sub-shelf melting, grounding line migration, and ice discharge have the potential to respond non-linearly to changes in ocean temperatures (**??????**), thus it is critical to take variability into account for quantitative assessments."

Technical corrections: Please also note the supplement to this comment: `http://www.the-cryosphere-discuss.net/tc-2017-47/tc-2017-47-RC3-supplement.pdf`

**References**

V. R. Barletta, L. S. Sørensen, and R. Forsberg. Scatter of mass changes estimates at basin scale for greenland and antarctica. *The Cryosphere*, 7:1411–1432, 2013. doi: 10.5194/tc-7-1411-2013.

Robert A. Bindschadler, Sophie Nowicki, Ayako Abe-Ouchi, Andy Aschwanden, Hyeungu Choi, Jim Fastook, Glen Granzow, Ralf Greve, Gail Gutowski, Ute Herzfeld, Charles Jackson, Jesse Johnson, Constantine Khroulev, Anders Levermann, William H. Lipscomb, Maria A. Martin, Mathieu Morlighem, Byron R. Parizek, David Pollard, Stephen F. Price, Diandong Ren, Fuyuki Saito, Tatsuru Sato, Hakime Seddik, Helene Seroussi, Kunio Takahashi, Ryan Walker, and Wei Li Wang. Ice-sheet model sensitivities to environmental forcing and their use in projecting future sea level (the searise project). *Journal of Glaciology*, 59(214):195–224, 2013. doi: doi:10.3189/2013JoG12J125.

J.A. Church, P.U. Clark, A. Cazenave, J.M. Gregory, S. Jevrejeva, A. Levermann, M.A. Merrifield, G.A. Milne, R.S. Nerem, P.D. Nunn, A.J. Payne, W.T. Pfeffer, D. Stammer, and A.S. Unnikrishnan. *Sea Level Change*, book section 13, pages 1137–1216. Cambridge University Press, Cambridge, United Kingdom and New York, NY, USA, 2013. ISBN ISBN 978-1-107-66182-0. doi: 10.1017/CBO9781107415324.026. URL www.climatechange2013.org.

L. Favier, G. Durand, S. L. Cornford, G. H. Gudmundsson, O. Gagliardini, F. Gillet-Chaulet, T. Zwinger, a. J. Payne, and a. M. Le Brocq. Retreat of Pine Island Glacier controlled by marine ice-sheet instability. *Nature Climate Change*, 4:117–121, 2014. doi: 10.1038/nclimate2094.

X. Fettweis, B. Franco, M. Tedesco, J. H. van Angelen, J. T. M. Lenaerts, M. R. van den Broeke, and H. Gallée. Estimating Greenland ice sheet surface mass balance contribution to future sea level rise using the regional atmospheric climate model MAR. *The Cryosphere*, 7:469–489, 2013. doi: 10.5194/tc-7-469-2013.

Christopher J. Fogwill, Christian S M Turney, Katrin J. Meissner, Nicholas R. Golledge, Paul Spence, Jason L. Roberts, Mathew H. England, Richard T. Jones, and Lionel Carter. Testing the sensitivity of the East Antarctic Ice Sheet to Southern Ocean dynamics: Past changes and future implications. *Journal of Quaternary Science*, 29(1):91–98, 2014. doi: 10.1002/jqs.2683.

Katja Frieler, Peter U. Clark, Feng He, Christo Buizert, Ronja Reese, Stefan R. M. Ligtenberg, Michiel R. van den Broeke, Ricarda Winkelmann, and Anders Levermann. Consistent evidence of increasing antarctic accumulation with warming. *Nature Climate Change*, 5:348–352, 2015. doi: 10.1038/NCLIMATE2574.

Jeremy G. Fyke, Miren, Vizcaíno, William Lipscomb, and Stephen Price. Future climate warming increases greenland ice sheet surface mass balance variability. *Geophysical Research Letters*, 41(2):470–475, 2014. doi: 10.1002/2013GL058172.

N. R. Golledge, D. E. Kowalewski, T. R. Naish, R. H. Levy, C. J. Fogwill, and E. G. W. Gasson. The multi-millennial antarctic commitment to future sea-level rise. *Nature*, 526:421–425, 2015. doi: doi:10.1038/nature15706.

I. Joughin, B. E. Smith, D. E. Shean, and D. Floricioiu. Brief communication: Further summer speedup of Jakobshavn Isbræ. *The Cryosphere*, 8(1):209–214, 2014. doi: 10.5194/tc-8-209-2014.

M. Mengel and A. Levermann. Ice plug prevents irreversible discharge from East Antarctica. *Nature Climate Change*, 4(6):451–455, 2014. doi: 10.1038/nclimate2226.

Ioana S. Muresan, Shfaqat A. Khan, Andy Aschwanden, Constantine Khroulev, Tonie Van Dam, Jonathan Bamber, Michiel R. van den Broeke, Bert Wouters, Peter Kuipers Munneke, and Kurt H. Kjær. Modelled glacier dynamics over the last quarter of a century at jakobshavn isbræ. *The Cryosphere*, 10(2): 597–611, 2016. doi: 10.5194/tc-10-597-2016.

Sophie M. J. Nowicki, Anthony Payne, Eric Larour, Helene Seroussi, Heiko Goelzer, , William Lipscomb, Jonathan Gregory, Ayako Abe-Ouchi, and Andrew Shepherd. Ice sheet model intercomparison project (ismip6) contribution to cmip6. *Geoscientific Model Development*, 9:4521–4545, 2016. doi: 10.5194/gmd-9-4521-2016.

David Pollard, Robert M. DeConto, and Richard B. Alley. Potential antarctic ice sheet retreat driven by hydrofracturing and ice cliff failure. *Earth and Planetary Science Letters*, 412:112 – 121, 2015. doi: 10.1016/j.epsl.2014.12.035.

Jeff Ridley, Jonathan M. Gregory, Philippe Huybrechts, and Jason Lowe. Thresholds for irreversible decline of the Greenland ice sheet. *Climate Dynamics*, 35(6):1049–1057, 2010. doi: 10.1007/s00382-009-0646-0.

Alexander Robinson, Reinhard Calov, and Andrey Ganopolski. Multistability and critical thresholds of the Greenland ice sheet. *Nature Climate Change*, 2 (6):429–432, 2012. doi: 10.1038/nclimate1449.

G. H. Roe and R. S. Lindzen. A one-dimensional model for the interaction between continental-scale ice sheets and atmospheric stationary waves. *Climate Dynamics*, 17(5–6):479–487, 2001. doi: https://doi.org/10.1007/s003820000123.

Gerard H. Roe and Michael O'Neal. The response of glaciers to intrinsic climate variability: observations and models of late-holocene variations in the pacific northwest. *Journal of Glaciology*, 55(193):839–854, 2005. doi: 10.3189/002214309790152438.

Julien Seguinot. Spatial and seasonal effects of temperature variability in a positive degree-day glacier surface mass-balance model. *Journal of Glaciology*, 59(218):1202–1204, 2013. doi: 10.3189/2013JoG13J081.

H. Seroussi, M. Morlighem, E. Rignot, J. Mouginot, E. Larour, M. Schodlok, and A. Khazendar. Sensitivity of the dynamics of Pine Island Glacier, West Antarctica, to climate forcing for the next 50 years. *The Cryosphere*, 8(5): 1699–1710, 2014. doi: 10.5194/tc-8-1699-2014.

Dirk van As, Andreas Bech Mikkelsen, Morten Holtegaard Nielsen, Jason E. Box, Lillemor Claesson Liljedahl, Katrin Lindbäck, Lincoln Pitcher, and Bent Hasholt. Hypsometric amplification and routing moderation of greenland ice sheet meltwater release. *The Cryosphere*, 11(3):1371–1386, 2017. doi: https://doi.org/10.5194/tc-2016-285.

J. Weertman. Stability of ice-age ice sheets. *Journal of Geophysical Research*, 66(11):3783–3792, 1961. doi: https://doi.org/10.1029/jz066i011p03783.

J. Weertman. Rate of growth or shrinkage of nonequilibrium ice sheets. *Journal of Glaciology*, 5(38):145–158, 1964. doi: https://doi.org/10.1017/s0022143000028744.

Johannes Weertman. Milankovitch solar radiation variations and ice age ice sheet sizes. *Nature*, 261(5555):17–20, 1976. doi: https://doi.org/10.1038/261017a0.

---

## Referee Report (RR1)

Review of revised manuscript "Influence of temperature fluctuations on equilibrium ice sheet volume" by Troels Bøgeholm Mikkelsen, Aslak Grinsted and Peter Ditlevsen [The Cryosphere Discuss. doi:10.5194/tc-2017-47

General comments
This manuscript has improved much from previous version and the context is clearer with the new abstract and introduction. The distinction that needs to be made between the minimal model (that authors state does not apply to Greenland ice sheet) and the estimates based on the results from Robinson is still not always clear, see comments below, particularly in the abstract (1m sle vs 30 GT/y) and in the conclusion „considering minimal model of the Greenland Ice sheet". It should be well separated and made clear when each model is applicable and what conclusions can be drawn – and what they mean. The reviewer Fettweis points out that the bias authors are pointing at is not of concern when the ice sheet models are forced with climate model output, this should also be stated in the paper so that readers will not be mislead to think that there is a bias in all large scale simulations of ice sheets. I have a few minor comments that could improve the text further.

Specific comments:
In the abstract it is not clear when authors are referring to the simple model (lines 4-7) and when to estimates based on simulations of Greenland Ice sheet (line 9) this should be clearly stated. For example by starting sentence in line 4 (We find) – by something like : By applying a simple circular symmetry model it is shown that steady state volume is biased toward a larger size if interannual temperature fluctuations are not taken into account, this can be approximately 1 m sea level equivalent for that setting. The text is confusing as it is now. The 1m sle is referring to the simple model, but the 30 GT is for Greenland, right?
Lines 4 and 7, suggest to replace "temporal" with "interannual"

Line 18  not clear what "full regional climate model" is, do you mean "high resolution"

Line 1 add "by" before solid
Line 2, is there a reference for this statement, or is this your concern? Then state that
Line 4, suggest to replace "response from" with "response to"
Line 5 – the sentence starting in line 5 needs editing, is this a result that you are presenting here or is there a reference you can use to support this statement?
Line 10, also here is a statement that needs supporting reference or clarification
Lines 17-19 this paragraph is misplaced, maybe it can be put into the previous paragraph, as it stands it is in no context with the rest of the section.
Line 20 if something is well known you should add some reference for the reader who is interested to learn more about this well known fact.
Line 24, take the plural s off models
Line 26, missing what the bias correction is applied to
Line 32 replace "an" with "a" long-term
Line 33 suggest to replace "differs" with "differ"  and "dependent" with "depending"

Line 3 add s to "influence"
Line 5 add "be" in front of applied  OR replace "applied" with "apply"
Line 6, something is missing, relationship between T and what?

Line 30, think it would be clearer to replace "mass balance" with "change in volume"

Figure caption - the "runoff line" is not explained and in this context it is not clear what the line is

Line 13, suggest to replace "for the model presented in Section 3" with Oer03 to clarify

Line 11, suggest to edit "For this value it is seen" – change to something like Figure 4 shows, or it can be seen on
Line 15 "Our results indicate…" This statement needs more clarification Figure 4. Shows that for 3°C warming the temperature bias is 0.12°C, how does that translate to the 1.6°C threshold for GrIS ice loss?
Line 16 ",but it reduces the window available to avoid passing this threshold" is a strange sentence, what do you mean here? That there is less temperature change needed to reach the threshold?
Line 18 Figure 4 shows the ΔSMB with units mm SLE yr-1, which here is translated to Gt/yr, suggest to use only one unit, or explain the assumption made to transfer from one to the other.
Line 31 to Page 10 line 6 - This paragraph is a strange way to start the conclusion section, suggest to move this to the discussion section

Line 5 this sentence is misleading and in contrast to the replies to previous comments, you state there that the minimal model, Oer03, is there to show how equation 6 works and that it is not to model Greenland Ice sheet. Here, however, you state that you have considered a minimal model of the Greenland Ice sheet, suggest to edit this sentence
Line 10-11, check the reference there should not be a parenthesis around the year within the parenthesis

The supplement for the paper is not very comprehensive and would benefit from a little more text to explain better its context. The text is very minimal and in bullet point style and does not provide the information needed to support the main text. The length of the main paper is not such that it would make it impossible to add the information in the supplement to the main text, making the paper more comprehensive and readable. The other option would be to provide more context in the supplement.
note that data is plural so line 3 should be ".. data consist of monthly means and are …"

page 8
line 2, suggest to replace "yearly" with "annual"
line 3, parameters… are used

page 9
see comment above the figure caption does not clarify what is going on here, more text would help putting the context clearer.
Same for page 10

---

## Author Response (AR2)

**Response to Interactive Comments**

Troels Bøgeholm Mikkelsen, Aslak Grinsted and Peter D. Ditlevsen

November 4, 2017

**1 Remark from the authors**

Dear editor and reviewers,

Once again we are grateful for your comments and suggestions, and for your valuable time spent reading the manuscript a second time.

We believe that we have now replied to all your comments and hope that this version of the manuscript is clearer and more precise than the previous version. Our replies to your comments and the changes made in the manuscript follow below.

Yours sincerely,
Troels Mikkelsen, Aslak Grinsted and Peter Ditlevsen

**2 Report #2 by Anonymous Referee #3**

**General comments**

> This manuscript has improved much from previous version and the context is clearer with the new abstract and introduction. The distinction that needs to be made between the minimal model (that authors state does not apply to Greenland ice sheet) and the estimates based on the results from Robinson is still not always clear, see comments below, particularly in the abstract (1m sle vs 30 GT/y) and in the conclusion "considering minimal model of the Greenland Ice sheet". It should be well separated and made clear when each model is applicable and what conclusions can be drawn – and what they mean.

**Comment #1:** Thank you for pointing this out.
**Change #1:** We have stated more explicitly in the abstract when the results relate to the simple model or the GrIS simulations by Robinson et al. (2012).

> The reviewer Fettweis points out that the bias authors are pointing at is not of concern when the ice sheet models are forced with climate model output, this should also be stated in the paper so that readers will not be mislead to think that there is a bias in all large scale

simulations of ice sheets. I have a few minor comments that could improve the text further.

**Comment #2:** Thank you for pointing this out.
**Change #2:** We now state in the conclusion "Temperature fluctuations may also be explicitly accounted for by forcing the ice sheet model with climate model output that reproduces the magnitude of observed interannual temperature variability."

**Specific comments**

**Page 1**

In the abstract it is not clear when authors are referring to the simple model (lines 4-7) and when to estimates based on simulations of Greenland Ice sheet (line 9) this should be clearly stated. For example by starting sentence in line 4 (We find) – by something like : By applying a simple circular symmetry model it is shown that steady state volume is biased toward a larger size if interannual temperature fluctuations are not taken into account, this can be approximately 1 m sea level equivalent for that setting. The text is confusing as it is now. The 1m sle is referring to the simple model, but the 30 GT is for Greenland, right?

**Comment #3:** Again, thanks for pointing out this imprecision. As per the comment above we have changed the wording in the abstract.

Lines 4 and 7, suggest to replace "temporal" with "interannual"

**Change #3:** Changed 'temporal' to 'interannual'

Line 18 not clear what "full regional climate model" is, do you mean "high resolution"

**Comment #4:** The word "full" here was meant to distinguish the regional model MAR used in Fettweis et al. (2013) from the simple models in the studies mentioned previously in that sentence. We already use the word "simple" to describe the simple models, and the word "full" is likely used in a non-standard way here.
**Change #4:** We have removed the word "full".

**Page 2**

Line 1 add "by" before solid

**Change #5:** Fixed.

Line 2, is there a reference for this statement, or is this your concern? Then state that

**Change #6:** We have added a reference and edited that sentence.

Line 4, suggest to replace "response from" with "response to"

**Change #7:** Fixed.

> Line 5 – the sentence starting in line 5 needs editing, is this a result that you are presenting here or is there a reference you can use to support this statement?

**Change #8:** We have changed the sentence to "Using a simple ice sheet model we will show how, as a consequence of this nonlinearity, the average mass balance will be different when forcing the model with a variable climate compared to a constant average climate."

> Line 10, also here is a statement that needs supporting reference or clarification

**Comment #5:** This statement is supported by the two references given in the previous sentence.

**Change #9:** We have edited the text so that this is clearer.

> Lines 17-19 this paragraph is misplaced, maybe it can be put into the previous paragraph, as it stands it is in no context with the rest of the section.

**Comment #6:** Thank you for pointing this out.

**Change #10:** We have moved the paragraph further down in the text where temperature variability is discussed further.

> Line 20 if something is well known you should add some reference for the reader who is interested to learn more about this well known fact.

**Comment #7:** Thank you for asking for clarification about this. "That the SMB of an ice sheet model is nonlinear with respect to temperature" is supported by the 3 references in the following sentences in that paragraph. However, that does of course not guarantee that the fact is "well known".

**Change #11:** We have changed "is well known" to "has previously been investigated in several studies".

> Line 24, take the plural s off models

**Comment #8:** Outputs of several models are compared in Fettweis et al. (2013, Fig 6.), so we believe that "models" should be plural. However we did write GCM and RCM as singular in the same sentence.

**Change #12:** Added plural s to GCM and RCM.

> Line 26, missing what the bias correction is applied to

**Change #13:** Added "to surface temperature".

> Line 32 replace "an" with "a" long-term

**Change #14:** Fixed.

> Line 33 suggest to replace "differs" with "differ" and "dependent" with "depending"

**Change #15:** Agree, fixed.

**Page 3**

Line 3 add s to "influence"

**Change #16:** Fixed.

Line 5 add "be" in front of applied OR replace "applied" with "apply"

**Change #17:** Fixed.

Line 6, something is missing, relationship between T and what?

**Comment #9:** A keen eye was required for spotting this.
**Change #18:** Changed to "relationship between the magnitude of temperature fluctuations and ice sheet volume"

Line 30, think it would be clearer to replace "mass balance" with "change in volume"

**Change #19:** Fixed.

**Page 4**

Figure caption - the "runoff line" is not explained and in this context it is not clear what the line is

**Comment #10:** Agree, this could have been written more clearly.
**Change #20:** Changed to "The runoff line $h_r$ specifies the simplified climatic conditions, as the specific balance is constant above $h_r$ (see also Supplementing Information, Eq. (4)), and the balance gradient is constant below $h_r$ (Oerlemans, 2003)'

**Page 5**

Line 13, suggest to replace "for the model presented in Section 3" with Oer03 to clarify

**Change #21:** Fixed.

**Page 9**

Line 11, suggest to edit "For this value it is seen" – change to something like Figure 4 shows, or it can be seen on

**Change #22:** Changed to "it can be seen on"

Line 15 "Our results indicate. . . " This statement needs more clarification Figure 4. Shows that for 3°C warming the temperature bias is 0.12°C, how does that translate to the 1.6°C threshold for GrIS ice loss?

**Comment #11:** As mentioned previously in the paragraph, if assuming RCP45 scenario the bias correction of 0.12 °C (0.10 °C – 0.18 °C) should be added to any constant warming threshold; this will shift the threshold estimate to colder temperatures.

**Change #23:** We have changed the text to clarify this, adding "Applying the bias correction above [...]"

Line 16 ",but it reduces the window available to avoid passing this threshold" is a strange sentence, what do you mean here? That there is less temperature change needed to reach the threshold?

**Change #24:** Changed to "but it places additional constraints on the maximum temperature increase admissible to avoid passing this threshold"

Line 18 Figure 4 shows the ΔSMB with units mm SLE yr-1, which here is translated to Gt/yr, suggest to use only one unit, or explain the assumption made to transfer from one to the other.

**Change #25:** Agree, we have added the details of the conversion to the supplementing information.

Line 31 to Page 10 line 6 - This paragraph is a strange way to start the conclusion section, suggest to move this to the discussion section

**Change #26:** That paragraph has been moved to the discussion. Additionally, the following paragraph has been moved down to improve the flow of the text.

**Page 11**

Line 5 this sentence is misleading and in contrast to the replies to previous comments, you state there that the minimal model, Oer03, is there to show how equation 6 works and that it is not to model Greenland Ice sheet. Here, however, you state that you have considered a minimal model of the Greenland Ice sheet, suggest to edit this sentence

**Comment #12:** Agree, this is an oversight on our part. Thank you for pointing this out.

**Change #27:** Removed reference to the Greenland ice sheet from that sentence.

Line 10-11, check the reference there should not be a parenthesis around the year within the parenthesis

**Change #28:** Fixed.

The supplement for the paper is not very comprehensive and would benefit from a little more text to explain better its context. The text is very minimal and in bullet point style and does not provide the information needed to support the main text. The length of the main paper is not such that it would make it impossible to add the information in the supplement to the main text, making the paper more comprehensive and readable. The other option would be to provide more context in the supplement.

**Comment #13:** Thank you for this comment. We have added more text to the supplement and hope that it is now comprehensive.
**Change #29:** Added text to the supplement.

**Page 7 (Supplement)**

> note that data is plural so line 3 should be ".. data consist of monthly means and are ..."

**Change #30:** Fixed.

**Page 8 (Supplement)**

> line 2, suggest to replace "yearly" with "annual"

**Change #31:** Fixed.

> line 3, parameters... are used

**Change #32:** Fixed.

**Page 9 (Supplement)**

> see comment above the figure caption does not clarify what is going on here, more text would help putting the context clearer. Same for page 10

**Comment #14:** Agree, as per the above comment we provide more context in the supplement.

**3 Report #3 by Gerard Roe**

> I've reviewed the revewers' comments and the authors' response. The authors have made comprehensive revisions in response to reviewer's suggestions. I think the analysis is worth publishing. I would recommend that the authors add a discussion of other sources of stochastic variability, so they can put their results in context. With 50% of Greenland ablation coming through calving, stochasticity in calving and in ice-stream dynamics are likely a source of stochastic forcing that will also have an impact on ice-sheet size. As a reader of the current manuscript, I'd want to know if the authors felt that it was just temperature variability I needed to incorporate or if there were potentially bigger problems out there.

**Comment #15:** Thank you for this comment, we are very happy to hear this. Regarding your point about other sources of stochastic variability, this is would undoubtedly lead to a very interesting discussion. Interesting work is being done in this field, such as that Mantellia et al. (2016) which we find interesting both in the context of ice sheets, as well as in the context of dynamical systems. As it stands now, in addition the mass balance response to to surface temperature variability, we discuss ablation-induced variability of the GrIS surface mass balance, the effects of ocean temperature on ice-discharge, and the case accumulation dominated mass balance, and feel that the addition of a further topic to the discussion would introduce too much material.

**Notes:**

-Abstract: increases in future variability are small if at all (Simolo et al., 2011, Rhines and Huybers, 2013). Does this claim get repeated anywhere else? I did not see it, so maybe just leave out.

**Change #33:** We have removed that sentence.

-Roe and ONeal is 2009, not 2005

**Change #34:** Fixed, thank you.

Simolo CM, Brunetti MM and Nanni T (2011) Evolution of extreme temperatures in a warming climate. Geophys. Res. Lett., 38(16), L16701

Rhines A and Huybers P (2013) Frequent summer temperature extremes reflect changes in the mean, not the variance. Proc. Natl. Acad. Sci., 110(7), E546–E546 (doi: 10.1073/pnas.1218748110)

[revised manuscript text omitted]